



# Atmospheric Blocking: The Impact of Topography in an Idealized General Circulation Model

Veeshan Narinesingh[1,2], James F. Booth[1,2], Spencer K. Clark[3], Yi Ming[4]

[1]Department of Physics, City University of New York – The Graduate Center, New York, New York, 10016, United States of America

[2]Department of Earth and Atmospheric Sciences and NOAA-CESSRST, City University of New York – City College, New York, New York, 10031, United States of America

[3]Program in Atmospheric and Oceanic Sciences, Princeton University, Princeton, New Jersey, 08544, United States of America

[4] Atmospheric Physics Division, NOAA Geophysical Fluid Dynamics Laboratory, Princeton, New Jersey, 08540, United States of America

*Correspondence to*: Veeshan Narinesingh (veenarinesingh@gmail.com)



**Abstract.**

Atmospheric blocking can have important impacts on weather hazards, but the fundamental dynamics of blocking are
not yet fully understood. As such, this work investigates the influence of topography on atmospheric blocking in terms of
dynamics, spatial frequency, duration and displacement. Using an idealized GCM, an aquaplanet integration, and integrations
with topography are analyzed. Block-centered composites show midlatitude aquaplanet blocks exhibit similar wave activity
flux behavior to those observed in reality, whereas high-latitude blocks do not. The addition of topography significantly
increases blocking and determines distinct regions where blocks are most likely to occur. These regions are found near high-
pressure anomalies in the stationary waves and near storm track exit regions. Focusing on block duration, blocks originating
near topography are found to last longer than those that are formed without or far from topography but have qualitatively
similar evolutions in terms of nearby geopotential height anomalies and wave activity fluxes in composites. Integrations with
two mountains have greater amounts of blocking compared to the single mountain case, however, the longitudinal spacing
between the mountains is important for how much blocking occurs. Comparison between integrations with longitudinally long
and short ocean basins show that more blocking occurs when storm track exits spatially overlap with high-pressure maxima in
stationary waves. These results have real-world implications, as they help explain the differences in blocking between the
Northern and Southern Hemisphere, and the differences between the Pacific and Atlantic regions in the Northern Hemisphere.





## 1 Introduction

Atmospheric blocks are quasi-stationary anticyclones that can cause temperature extremes (Sillman et al., 2011; Pfahl and Wernli, 2012), steer hurricanes and extratropical cyclones (Mattingly et al., 2015; Booth et al. 2017, respectively), and induce persistent weather (Cassou et al., 2005; Dole et al., 2011; Brunner et al., 2018). Despite the expensive and sometimes deadly impacts of blocks, many fundamental questions remain regarding their behavior, and models tend to underpredict blocks in terms of their frequency and duration (D'andrea et al., 1998; Matsueda, 2009). Wintertime blocks are particularly interesting, because they occur during the season when the jet stream and extratropical cyclones are strongest. As such, this paper seeks to expand our understanding of wintertime blocks, focusing on their dynamics, spatial distribution, frequency, duration, and displacement.

The climatological spatial distribution of blocks is well documented. In Winter for the Northern Hemisphere, two main regions of blocking occur at the north-eastern edges of the Atlantic and Pacific Ocean basins (Barriopedro et al., 2006; Croci-Maspoli et al., 2007; Dunn-Sigouin et al., 2013). In the Southern Hemisphere (SH), one main region of blocking exists, located southwest of South America (Renwick, 2005; Parsons et al., 2016; Brunner and Steiner, 2017). Overall, blocking occurs more frequently in the Northern Hemisphere (NH) than the Southern. This difference in blocking frequency is likely related to the stronger stationary wave in the NH, often attributed to more prominent midlatitude topography and land-sea contrasts in the NH, e.g., Held et al. (2002). However, to our knowledge, no study has confirmed this assumption.

Why blocks preferentially occur in certain regions remains unclear. Some have argued that blocks are consequences of an interaction between eddies and stationary waves induced by orography (Egger, 1978; Charney and Devore, 1979; Tung and Lindzen, 1979; Luo, 2005). These studies suggest mountains are critical for setting the location of climatological block maxima. On the other hand, Shutts (1983) uses an idealized model to show that blocking flows do not necessarily need stationary forcing and can arise purely through interactions between transient eddies. This suggests the extratropical cyclones (i.e., synoptic-scale eddies) that occur upstream of the blocking regions may be key. Related to this Colucci (1985) and Pfahl et al. (2015) show that extratropical cyclones can impact blocks downstream of the storm track exit region. In a related theory, blocks are linked to Rossby wave-breaking (Pelly and Hoskins, 2003; Berrisford, 2007; Masato et al., 2012), and wave-breaking occurs more frequently at the storm track exit regions. Thus, for the NH, there are two factors that might have important roles in determining the characteristics of blocking: the topographically induced stationary waves and the storm track exit regions.

The proposed factors that may influence blocking in the NH, make the presence of blocking in the SH all the more interesting. Most SH blocks occur at higher latitudes than the NH counterparts, and have less impact on the zonal flow (Berrisford et al., 2007). However, blocks can occur throughout the SH storm track region, far from topography. Related to this Hu et al. (2008) and Hassanzadeh et al. (2014) show blocks can occur in idealized aquaplanet configurations. Hu et al. (2008) find that blocks in their aquaplanet model occur more frequently than what is observed in nature – regardless of hemisphere, which is contradictory to the idea that stationary waves facilitate blocking episodes. The results of Hu et al. (2008)



however, are complicated by known discrepancies within the community regarding the identification of blocks (e.g. Barnes et
al., 2012), where they compare their idealized model results to a study that uses an alternate block identification metric on
reanalysis data. Thus, questions remain regarding the relatively frequency of blocks with and without the presence of
topography.
The work herein focuses on climatological and dynamical aspects of atmospheric blocks using an idealized moist
GCM. The first analysis focuses on blocks in an aquaplanet configuration.   For this, we present composites of the evolution
of geopotential height anomalies and wave activity flux throughout block lifecycles and compare midlatitude and high-latitude
blocking. For the second analysis, we start by studying the climatological flow features and block spatial frequencies in
reanalysis as a benchmark. We then add topography to the aquaplanet and examine the response of climatological flow features
and block spatial frequency. We adjust the number and placement of the topographic features so that we can determine the
response of blocking to topography, the storm tracks, and the distance between the two features, i.e., the zonal extent of the
ocean basins between the topographical features. Finally, the third analysis examines the sensitivity of block duration and
displacement for the different topographical configurations.

**2 Methods**
**2.1 Reanalysis data**
Although the focus of this paper is on a set of idealized numerical modelling experiments, we first present results using
reanalysis to motivate our work. The reanalysis used is the ECMWF ERA-Interim dataset (Dee et al., 2011). ERA-Interim
(ERAI) has been shown to represent winter midlatitude storms as well as, and in some cases better than, other reanalyses
(Hodges et al., 2011). Therefore, it likely does a reasonable job at capturing atmospheric blocking. ERA-Interim is produced
using a model with roughly 0.67-degree resolution, but it is available to download at different resolutions. Herein, we used
data with a 1.5 x 1.5 degree horizontal resolution. For this analysis we focus only on winter, which is defined as Dec. – Feb.
(DJF) and Jun. – Aug. (JJA) for the Northern and Southern Hemispheres, respectively.

**2.2 Idealized model configuration**
This work utilizes an idealized moist GCM described by Clark et al. (2018; 2019), which is modified from that
introduced by Frierson et. al. (2006; 2007) and later altered by Frierson (2007) and O'Gorman and Schneider (2008). The
model is configured to use 30 unevenly spaced vertical sigma coordinate levels, and T42 spectral resolution, corresponding to
64 latitude by 128 longitude grid points when transformed to a latitude-longitude grid. Earth-like orbital parameters are used
to simulate a full seasonal cycle in solar insolation. The model includes full radiative transfer and simplified physics
parameterizations of convection (Frierson, 2007), boundary layer turbulence (Troen and Mahrt, 1986), and surface fluxes.
There is no treatment of cloud radiative effects or condensed water in the atmosphere.
An aquaplanet configuration is run as the control integration. In other integrations, configurations of topographical
forcing are simulated by modifying the model surface height and using a simplified treatment of land following Geen et al.



(2017) and Vallis et al. (2018). Like Cook and Held (1992), and Lutsko and Held (2016), perturbations to the surface height
are introduced in the form of Gaussian mountains centered at 45° N with half-widths of 15 degrees in both the latitude and
longitude dimensions. Here, mountains are placed in various zonal configurations for the topographic integrations (Figure 1).
The 4 configurations are:

   a)   Aquaplanet

b)   SingleMtn: Single 3 km high Gaussian mountain centered at 45° N, 90° E.
c)   SymMtn: Two Symmetrically placed 3 km high Gaussian mountains centered at 45° N, 90° E and 45° N, 90° W

respectively.

d)   AsymMtn: Two Asymmetrically placed 3 km high Gaussian mountains centered at 45° N, 90° E and 45° N, 150° W

respectively.

Ocean grid cells contain a 20-m slab ocean, and as a simplification, are assumed to redistribute zero energy

horizontally; that is, we prescribe uniformly zero heat flux, often referred to as a Q-flux, into or out of the ocean. In the
configurations with topography, land grid cells are defined as locations where the height is greater than 1/100th of the
maximum surface height (3 km), corresponding to a height threshold of 30 m. As in Geen et al. (2017) and Vallis et al. (2018)
land is simulated by reducing the slab ocean depth to 2 m (effectively reducing the heat capacity) and limiting evaporation
using a bucket hydrology model. A uniform surface albedo of 0.26 is used to obtain a global annual mean surface temperature
resembling that of the Earth. Each configuration is integrated for 40 years, but the first 10 years are discarded as spin-up time.
Thus, the results presented here are for years 11-40 of each integration. 6-hourly data sets are used for the analyses in this
paper, and the results are presented for Northern Hemisphere Winter, defined as the 3 months centered on the minimum in
solar insolation. The model data is interpolated to the 1.5 x 1.5 degree horizontal ERA-Interim resolution prior to any analysis.

**2.3 Block detection and tracking**

Here we use a 500 hPa geopotential height (Z500) hybrid metric that utilizes the Z500 anomaly and meridional

gradient. This metric was chosen for its robustness   in terms of capturing high amplitude events involving wave-breaking
(Dunn-Sigouin et al., 2013), and because it only requires the Z500 field – which simplifies tracking when analyzing large
datasets. Barnes et al. (2012) finds that utilizing a Z500 metric produces similar blocking durations and climatologies to both
potential vorticity and potential temperature based metrics. Blocks are detected and tracked using the algorithm described by
Dunn-Sigouin et. al. (2013), hereinafter as DS13, which is an adaptation of previous methods by Barriopedro et al. (2010) and
Sausen et al. (1995). This algorithm searches for large, contiguous regions of persistent, high amplitude, positive anomalies in
the Z500 field. Within these regions, the Z500 must satisfy a meridional gradient reversal condition. What follows is an
overview of the block identification algorithm, but specific details can be found in DS13:
1.   Z500 Anomaly Calculation: For each grid-point poleward of 30 N, from the raw Z500 field subtract the running

annual mean and mean seasonal cycle as computed in DS13.



2.  Normalize each anomaly value by the sin of its latitude divided by sin of 45 degrees, i.e. $\frac{\sin(\phi_{ij})}{\sin(45°)}$, where $\phi_{ij}$ is the

latitude of an arbitrary grid-point with longitude $i$ and latitude $j$. This normalized anomaly will be referred to as Z500'.

3.  For each month, in a 3-month window centered on a given month, calculate the standard deviation, S, of all Z500'

values.

4.  Amplitude threshold: Identify contiguous regions of positive Z500' greater or equal to 1.5*S.
5.  Size threshold: Regions must be at least 2.5 x $10^6$ km$^2$ in area.
6.  Gradient Reversal: The meridional gradient of the Z500 field within candidate regions must undergo a reversal in

sign as described by DS13.

7.  Quasi-stationary condition: For each timestep, regions must have a 50 % area overlap with its previous timestep

(modified from DS13's 2 day overlap which was applied to daily mean data)

8.  Blocks must meet the above criteria for at least 5 days (e.g. 20 6-hourly timesteps)

In case studies using ERAI and the idealized configurations described here, it was observed that two existing blocks

sometimes merged with one another to form a single, larger block. We objectively identified this merging process based on
extreme shifts in the location of the block centroid (defined as the gridpoint that is the centroid of the anomalous area associated
with the block). If the centroid shifted by more than 1500 km from one 6-hourly snapshot to the next, we labeled the block as
a merged event. These merged events represented 23-27 percent of the total initial blocks found in the four configurations. We
judge these events to be unique in terms of their relationship between block duration. Furthermore, the merger-blocks create
uncertainty in terms of defining a block center for the sake of our block-centered composite analysis. Therefore, we have
excluded the merged events from our analysis, and plan future work focused on these merged-block events.

**2.4 Analysis metrics**

The metrics used to characterize climatological features and blocking in the idealized model data and reanalysis are

outlined below.

**2.4.1 Stationary wave**

The winter stationary wave at each point is defined as the anomaly with respect to the zonal mean of the winter

climatology for the 250-hPa geopotential height field: $\bar{Z}^* = \bar{Z} - \left[\bar{Z}\right]$, where brackets indicate the zonal mean and overbar
indicates the time mean over winter days for all years. This is computed separately for each gridpoint.

**2.4.2 250 hPa zonal wind climatology**

The 250 hPa zonal wind climatology (U250) is presented as the time mean of the 250-hPa zonal wind over the winter

months at each gridpoint.





### 2.4.3 Eulerian storm track

The storm track is presented as the standard deviation of a 24-hour difference of the daily mean Z500 field during winter (Wallace et all. 1988). Consider $Z_{500}(t)$ to be the daily mean Z500 value for an arbitrary gridpoint. To obtain the storm track:

1.  The 24-hour difference, $Z_{500}^{\tau}$, at each gridpoint is taken as:

$$Z_{500}^{\tau} = Z_{500}(t+1) - Z_{500}(t)$$

2.  Then, the standard deviation of $Z_{500}^{\tau}$ for all winter timesteps at each gridpoint is taken to obtain the winter Eulerian storm track value at that point.

This is computed separately for each gridpoint.

### 2.4.4 Blocking climatology

Blocking climatologies are calculated by averaging the block identification flag (1 or 0 respectively) per gridpoint over all winter days. Thus, the blocking climatologies show the percent of winter timesteps a block (as defined here) is present. This is computed separately at each gridpoint.

### 2.4.5 Wave activity flux vectors

To better characterize the dynamical evolution of blocks within each model, wave activity flux vectors (hereinafter, $\vec{W}$) are calculated as described by Takaya and Nakamura (2001), hereinafter TN01. Block centered composites (as described in Sect. 2.5.1. of this paper) are then computed using $\vec{W}$ for each block during stages of the block's lifecycle. The horizontal components of $\vec{W}$ are calculated as in TN01. For this, only the stationary term was considered, which yielded similar results in reanalysis to those presented in the original TN01 article. $\vec{W}$ is given by:

$$\vec{W} = \frac{p\cos\phi}{2|\vec{U}|} \begin{pmatrix} U\left(v'^2 - \frac{\Phi'}{f}\frac{\partial v'}{\partial x}\right) + V\left(-u'v' + \frac{\Phi'}{f}\frac{\partial u'}{\partial x}\right) \\ U\left(-u'v' + \frac{\Phi'}{f}\frac{\partial v'}{\partial y}\right) + V(u'^2 + \frac{\Phi'}{f}\frac{\partial u'}{\partial y}) \end{pmatrix}$$

This calculation is performed on 250-hPa fields, where for each point, $p$ is the pressure, and $\phi$ is latitude. $\vec{U}$ is the 30-day low-pass filtered horizontal wind vector with zonal and meridional components $U$ and $V$, respectively. The 3-to-30-day bandpass filtered zonal wind, meridional wind, and geopotential are given by $u', v',$ and $\Phi'$, respectively. Derivatives are computed using finite-differencing, where zonal derivatives are weighted by latitude. $\vec{W}$ are given in units of $m^2s^{-2}$



## 2.5 Analysis methods

### 2.5.1 Block-centered compositing

The Z500' field and $\vec{W}$ are composited around the centroid of each block for the first, strongest, and final days of each block lifecycle per run. To account for the convergence of meridians, the Z500' field and $\vec{W}$ are projected onto equal-area grids before compositing. The initial time step of a block is the first timestep that the block satisfies the amplitude, size and reversal conditions. The strongest time step of a block is defined as the time step with the greatest Z500' (at a single lat/lon location) within a block. The final timestep is the last timestep a block satisfies the amplitude, size and reversal conditions.

The composites presented in this paper for the aquaplanet, unless otherwise stated, only include midlatitude-blocks whose centroid are always south of 65˚ N. This is because we find that the high-latitude blocks exhibit distinct physical behavior. The aquaplanet showed a greater tendency to produce more poleward blocks compared to the other configurations. From reanalysis data, high-latitude blocks in the Southern Hemisphere have different dynamical evolution and different impacts on the surrounding flow, as compared to midlatitude blocks (Berrisford et al., 2007). Based on these previous results and our own findings, we present separate results for the midlatitude (count = 95; see Sect. 3.1) and high-latitude blocks from the aquaplanet configuration (count = 46; see Sect. 3.2). The 65˚ N cutoff was chosen after estimates showed this to be near the minimum in the meridional potential vorticity gradient, and thus the northern limit of the midlatitude waveguide (e.g. Wirth et al. 2018). After changing the cutoff by +/- 5˚, 65˚ N proved to be the best compromise between distinguishing dynamical features between mid and high-latitude blocking, but also retaining enough members of each midlatitude and high-latitude subset.

### 2.5.2 Separating blocks by region

To compare the dynamical evolution of blocks originating near the eastern edge of the ocean basins (denoted as East, near the windward side of mountains and the high-pressure maxima of stationary waves) against blocks originating near middle of the ocean (denoted as Mid, near the end of the storm tracks), blocks are sorted by their centroid location during their first timestep. Each region spans 30˚-75˚ N for 100 degrees of longitude. In SymMtn, we defined our East region relative to the mountain at 90E, which behaved similarly in our analyses to a region defined by the 90 W mountain instead. For the AsymMtn configuration, East and Mid refer to two regions within the zonally larger ocean basin (which we refer to as the wide basin), whereas blocks originating within the other ocean basin are only denoted by short basin. These regions are summarized in Table 1.

### 2.5.3 Block duration probability density distributions

Block duration is defined as the time interval from the initial identification timestep to the end of that block's existence – based on the block identification algorithm (described in Sect. 2.3). Each block is thus assigned one duration value. The steps taken to obtain block duration probability density distributions are as follows:





1. Sort blocks into subsets by model configuration and/or basin.

2. Allowing replacement, randomly select a set of block durations within a given subset. The size of the random set is given by the number of blocks in the subset being analyzed.

3. Place the durations yielded by step 2 into n equal sized bins (n=8 for figures in this paper) ranging from the minimum to maximum duration of winter blocks between all model configurations.

4. Steps 2 and 3 are then repeated m times (m=1000 for figures in this paper) to produce an ensemble of m probability density distributions for each subset.

5. For a given subset, the mean probability density distribution is computed by taking the mean of that subset's distributions. This is then smoothed using a running mean.

6. For a given subset, the standard deviation of probability density distribution is computed by taking the standard deviation of that subset's distributions

The results of this paper are nearly constant with respect to changes in the values of n (+/- 2) and m (+/- 200). For all configurations, distributions and mean values presented for duration exclude any high-latitude blocking (blocks whose centroid are ever poleward of 65° N). 65° N was found to be the most appropriate cut-off in each configuration for the same reasons as described for the aquaplanet compositing.

**2.5.4 Block displacement**

To measure the propensity for individual blocks to move horizontally, we define a block displacement metric. In this metric for an arbitrary block, the great circle distance between the block centroid at successive timesteps is computed. The block displacement for each block is the sum of all displacements computed throughout its lifecycle, divided by the number of timesteps (i.e. the average centroid displacement every 6 hours).

**2.5.5 Statistical significance**

To compare block frequency between configurations, the area-weighted mean of winter block frequency is computed for each year of a given configuration. A 2-sample t-test is used on the yearly area-weighted mean values between configurations to test for significant differences. Between configurations and regions, significance testing discerning mean block duration and displacement employ a 2-sample t-test. An 85% confidence interval is imposed as the significance threshold for all significance testing.

**3 Results**

**3.1 Blocking in the aquaplanet**

According to our tracking algorithm, there are blocking events in the aquaplanet integration, which agrees with previous idealized modeling work (Hu et al., 2008; Hassanzadeh et al., 2014). An example of the beginning of a blocking episode in the aquaplanet can be seen in Figure 2. Upstream and coincident with the block, a Rossby wave pattern can be





observed in both the Z500 and Z500' fields (Fig. 2 - the Z500 contours show a wave-like feature, and the Z500' field shows
an alternating pattern of low and high anomalies in the zonal direction). The presence of these features during the formation
of a block agrees with previous work for both simplified (Berggren et al., 1949; Rex, 1950; Colucci, 1985; Nakamura et al.,
1997; Hu et al., 2008), and comprehensive models (TN01; Yamazaki and Itoh, 2013; Nakamura and Huang, 2018; Dong et
al., 2019). Concentrated, large magnitude $\vec{W}$ are found just upstream of, and propagating into the block, and a relative absence
of large magnitude $\vec{W}$ occur downstream of the block (Fig. 2). The behavior of $\vec{W}$ during the genesis of this block case study
agrees with Nakamura et al. (1997) and TN01.

We use block-centered compositing analysis to confirm that, on average, the blocks identified in the aquaplanet model

evolve in a dynamically similar manner to results shown in previous studies. Figure 3 shows block centered composites of
Z500' and $\vec{W}$ for the aquaplanet blocks in the midlatitudes (i.e., occurring between 30˚ and 65˚ of latitude). The onset of
blocking in the composite is similar to that found in the case study (Fig. 2): a Rossby wave train with low-pressure centers
upstream and downstream of the composite block centroid, and a large concentration of $\vec{W}$ upstream and entering the block
(Fig. 3a). For composites over blocks at maximum strength, the wave pattern is no longer pronounced and low pressure is
concentrated equatorward and downstream of the block (Fig. 3b). Large magnitude $\vec{W}$ are concentrated inside the block during
this time (Fig. 3b). On the final day, the composite block's Z500 anomaly weakens, and low-pressure is concentrated
downstream from the block (Fig. 3c). Weak values of $\vec{W}$ exit the block downstream of the high-pressure maximum during this
time (Fig. 3c). The composites shown here for the aquaplanet are qualitatively similar to composites for the model
configurations with topography, in terms of the evolution of the Z500' field and $\vec{W}$.

These compositing results for the midlatitude blocks in the aquaplanet are similar to previous results from reanalysis

(Nakamura et al., 1997; TN01; Nakamura and Huang, 2008) in that: (1) An envelope of maximum $\vec{W}$ moves from upstream,
to inside the block when it is at its maximum strength, to downstream of the block as the block decays (i.e., Fig. 3a-c), and,
(2) The geopotential height field shows the evolution of a wave train that eventually dissipates as $\vec{W}$ are passed downstream
(Fig. 3a-c). On the other hand, the high-latitude blocks from the aquaplanet display quite different behavior.

**3.2 High-latitude blocking**

As discussed in the methods section, the aquaplanet configuration has a larger amount of high-latitude blocking than

the other configurations (Fig. 4a, Table 2). These blocks have multiple unique characteristics, as compared to blocks from all
model configurations (including the midlatitude blocks for aquaplanet). Berrisford et al. (2007) report that high-latitude
blocking events in the Southern Hemisphere have unique behavior compared to their midlatitude counterparts, e.g., not
blocking westerly flow or transient eddies. With this as motivation, we present a separate block-centered analysis of the high-
latitude blocks from the aquaplanet integration.



Figure 4b shows the block centered composite of high latitude aquaplanet blocks during their strongest timestep.

High-latitude blocks primarily occur poleward of the primary latitudes of wave activity and synoptic systems. Compared to
the midlatitude blocks, the high-latitude blocks do not contain much of any large magnitude $\vec{W}$ during their strongest timestep
(Figs. 3b and 4b). Nakamura et al. (1997) and TN01 cite $\vec{W}$ as an important ingredient in block maintenance, but perhaps this
is not so true for high-latitude blocking episodes. Furthermore, the composite of high-latitude blocks has much lower
geopotential height anomalies than the midlatitude block composite. This unique behavior of high-latitude blocks in the
aquaplanet is consistent with that reported in Berrisford et al. (2007) for high-latitude blocks in the SH.

High-latitude blocking was also identified in the model configurations with topography, but with lesser frequency

(Fig. 4a). Composites comparing high-latitude to midlatitude blocks for each configuration yielded similar results to the
aquaplanet (not shown).

Overall, case studies and block-centered composites show that blocks in the idealized model share similar

characteristics and dynamics as blocks observed in reality. This holds even when blocks are sorted between midlatitude and
high-latitude events. Confident with the representation of blocking in the idealized model, next we focus on the climatological
response of blocking to topography.

**301   3.3 The effects of topography on winter blocking**

This section focuses on the effect of topography on climatological flow features and blocking climatologies. As

motivation, we first present results from reanalysis that agree with previously published studies. Then, we investigate the
response of the same climatological features in the idealized model to changes in topography.

**306   3.3.1 Reanalysis**

The different topographic configurations of the northern and southern hemispheres produce distinct spatial

distributions of general circulation features and atmospheric blocking. Figure 5 shows the stationary wave, U250 climatology,
storm track and blocking climatology for winter in ERA-Interim. Stationary wave patterns can emerge due to land-sea heating
contrasts and flow deflection by orographic geometry. The two strongest regions of anomalous high-pressure in the Northern
Hemisphere (NH) are located on the windward side of the Rocky Mountains, and near the western edge of Europe (Fig. 5a).
The high near the Rockies is part of a wave train induced by the mountains (e.g., White et al., 2017). The high near Europe is
more likely driven by land-sea contrast. The Asian orography also produces a stationary wave response and is an important
ingredient for the Pacific jet and storm track (Brayshaw et al., 2009). These results agree with previous studies (Valdes and
Hoskins, 1991; Held, 2002; White et al., 2017).

U250 in the NH has two distinct maxima situated over the eastern coastlines of North America and Asia (Fig. 5a.).

These maxima are downstream of the topography. These regions of maximal U250 play a key role in guiding storms and
creating storm tracks. The storm tracks are regions where transient eddies are most prevalent in the extratropics (e.g., Trenberth,





1991; Chang et al., 2002). The Northern Hemisphere storm tracks maximize just upstream of the U250 maxima (Figs. 5a and 5c). At the end of storm tracks, Rossby waves tend to break more frequently (Abatzoglou and Magnusdottir, 2006) which is often associated with blocking (Pelly and Hoskins, 2003; Masato et al., 2012).

The Northern Hemisphere blocking climatology agrees with previous work (Wallace et al., 1988; Barriopedro et al., 2006; Dunn-Sigouin, 2013). In the Pacific basin of the Northern Hemisphere, the spatial maximum in climatological block frequency (blocking maximum) is nearly co-located with the high-pressure anomaly of the stationary wave induced by the Rocky Mountains (Fig. 5a and Fig. 5c). This region is also spatially overlapping with the Pacific storm track exit. For the NH Atlantic basin, the location of the blocking maximum and high-pressure stationary maximum are within close proximity, but both the storm track exit *and* maximum spatially overlap with them (Fig. 5a and Fig. 5c). In the NH, blocks rarely occur near the low-pressure anomalies of the stationary wave (Fig. 5a and Fig. 5c).

In the Southern Hemisphere (SH), the high-pressure maximum is more poleward than the Northern Hemisphere maxima and stretches from the southwestern tip of South America into a secondary maximum southeast of Australia (Fig 5b). This matches what is reported in Quintanar and Mechoso (1995). Stationary wave features are far less apparent in the Southern Hemisphere, presumably because of the relative lack of topographic forcing compared to the Northern Hemisphere.

The lack of topographic forcing in the SH allows there to be one distinct band of maximum U250 (Fig 5b). The U250 maximum in the SH stretches from the Indian Ocean into the Pacific and maximizes East of Australia (Fig 5b). The single storm track maximizes in the Indian Ocean near Antarctica and stretches from the Atlantic to the Pacific (Fig 5d), far upstream from the region of maximum U250 (Fig 5b and Fig 5d). The SH storm track is as reported in Nakamura and Shimpo (2004).

In the Southern Hemisphere, our blocking climatology is similar to that reported in Brunner and Steiner (2017). The blocking maximum is near the high-pressure anomaly of the stationary wave and the exit region of the Pacific storm track of the Southern Ocean (Fig. 5b and Fig. 5d). The spatial frequency of blocking in the SH extends into the SH Atlantic storm track entrance region, away from the high-pressure anomaly, but the local blocking maxima in the SH Atlantic is weak compared to the SH Pacific maxima (Fig. 5d).

Topographic differences yield contrasting spatial distributions of stationary waves, U250, storm tracks, and blocking between the hemispheres. These observations lead to the specific questions this subsection seeks to address:

- What effect does topography have on blocking?
- What role do stationary-waves and storm track exit regions have in setting the locations and intensity of blocking maxima?

### 3.3.2 Blocking in idealized model experiments

The idealized model configurations allow us to systematically investigate the response of atmospheric circulation and blocking to topography. As we did for reanalysis, for each model configuration we examine the stationary wave, U250, storm tracks, and the blocking climatology.

*Stationary wave*





As expected, a stationary wave is absent in the aquaplanet (Fig. 6a), and upon introducing topography, zonally
asymmetric forcing is imposed, and a stationary wave is induced (Figs. 6b-6d). SingleMtn contains a high-pressure anomaly
near the coastline on the windward side of the mountain, and a low-pressure anomaly on the leeward side (Fig. 6b). This results
in a meridionally tilted stationary wave pattern that extends into the subtropics leeward of the mountain. This pattern has been
explained in previous idealized modeling work (Grose and Hoskins, 1979; Cook and Held, 1992; Lutsko 2016). The high-
pressure anomaly extends approximately 180° of longitude upstream of the mountain and weakens from east to west.
In SymMtn, the configuration with two mountains and equal-sized ocean basins, each mountain induces a
meridionally tilted stationary wave pattern (Fig. 6c) similar to that in SingleMtn (Fig. 6b). The zonal extent of the high- and
low-pressure anomalies in the SymMtn stationary waves, however, are suppressed compared to SingleMtn. This suppression
is due to interference of stationary waves induced by multiple sources of topographic forcing (Manabe and Terpstra, 1974;
Held et al., 2002; White et al., 2017).
For AsymMtn, the placement of the topography creates two ocean basins of different zonal extents: a short basin and
a wide basin. Like SymMtn, each mountain in AsymMtn induces a meridionally tilted stationary wave, however, the
asymmetric configuration of the mountains results in asymmetric zonal extent in the anomalies (Fig. 6d). In the short basin,
the anomalies have less zonal extent than those in SymMtn, and the opposite holds true for the wider basin. Further comparing
the two basins, we find the high-pressure anomaly in the wide basin extends 100 degrees westward from the mountain, much
farther than that of the short basin. This extended high-pressure anomaly is related to blocking, which we will further address
below, but first we analyze U250.

### 250 hPa zonal wind climatology

In the aquaplanet, U250 is zonally symmetric. When topography is added, localized regions of U250 maxima occur.
In SingleMtn, the U250 maximum occurs on the leeward side of the mountain, equatorward of the low-pressure anomaly (Fig.
6b). The stationary wave pattern associated with the topography generates cold advection towards the southeast on the lee of
the mountain. This is due to both the change in wind direction created by the mountain and the differences in heat capacity for
the topography as compared to the ocean. The cold advection leads to a local maximum in meridional temperature gradient
east of the topographical feature (not shown). Related to this temperature gradient, the U250 maximum must exist due to
thermal wind balance. This pattern of the U250 maximum occurring just downstream of mountains is the same as what occurs
for the NH in observations (Fig. 5a).
In SingleMtn there is also a relative suppression of U250 nearly 120° downstream of the mountain, from about 150°
W – 110° W, followed by a secondary maximum of U250 from roughly 110° W - 0°. The disjointed distribution of U250 is
perhaps a consequence of blocking and will be discussed further in the blocking climatology subsection.
The U250 maxima for SymMtn and AsymMtn are located on the poleward side of the low-pressure anomalies of each
configuration's stationary waves (Figs. 6c-6d). This is due to the same cold advection-generated temperature gradient
explained for SingleMtn. In AsymMtn Short Basin however, the zonal extent of the U250 maximum produced by the upstream



mountain is suppressed – likely because of influence from the downstream mountain, and consistent with the zonal suppression
of the stationary wave (Fig. 6d).

The U250 field acts as a waveguide for synoptic scale Rossby Waves -e.g. Wirth et al. (2018). The waveguide
coincides with preferred regions where transient Rossby Waves are generated and propagate. These regions are also known as
storm tracks (e.g. Chang et al., 2002), which is the next topic of our discussion.

*Eulerian storm track*

The storm track in the aquaplanet is zonally symmetric (Fig. 6e), while the topographical configurations have zonally
asymmetric storm tracks whose locations are set by mountains (Figs 6f-6h). In the topographic configurations, the storm tracks
almost exactly overlap with the U250 maxima, with the exception that the storm track maxima are slightly upstream from the
U250 maxima (Figs 6b-6d, Figs 6f-6h). In AsymMtn Short Basin, the zonal extent of the storm track is suppressed by the
topographical spacing, similar to U250 and the stationary wave.

Previous studies have shown that the storm track exit region coincides with the terminus of the midlatitude wave
guide (e.g. Wirth et al., 2018), and the exit region is the primary region where Rossby wave breaking occurs (Strong and
Magnusdottir, 2008; Davini et al., 2012). The storm track exit region can interact with the stationary wave, and as discussed
in the next section, the storm track exit proves to be very important in where and how frequently blocks occur.

*Blocking climatology*

The blocking climatology in the aquaplanet is not zonally symmetric for the 30-year integration (years 11-40; Fig.
6e). For a 300-year integration, the climatology is much closer to being zonally symmetric, though it has still not converged
(not shown). No zonal asymmetries in forcing exist in the aquaplanet, so the zonal asymmetries attest to the internal variability
and relative rarity of blocking events identified in the aquaplanet. The configurations with topography are closer to reaching
convergence after 30 years – in terms of the local maxima occurring just west of the topography. To demonstrate this, we
compare climatologies of Aquaplanet and AsymMtn based on randomly chosen subsets of years. Aquaplanet blocking
climatologies using randomly sampled years produce results with varying spatial frequencies (Fig. 7).

Upon adding topography, spatial maxima form in the blocking climatology (Figs. 6e-6h) and significantly more
blocking occurs overall (see Table 2 for quantitative differences). The result that adding topography to our model leads to
greater block frequency matches with observations, since the Northern Hemisphere contains a relative abundance of
topography and blocking, when compared to the Southern Hemisphere (Figs. 5c-5d).

The blocking maximum in SingleMtn (Fig. 6f) is slightly upstream from the maximum high-pressure anomaly (Fig.
6b), on the windward side of the topography. This is similar to the NH Pacific blocking maximum being situated northwest of
the Rocky Mountains in observations (Fig. 5c). The high-pressure anomaly on the windward sides of mountains acts as a
source region of anticyclonic vorticity and can be recognized as ridges in instantaneous maps of geopotential height. These



ridges serve as precursors for topographically induced blocks, which are then amplified and maintained by transient eddies
and $\vec{W}$ (Nakamura et al., 1997; TN01).
A secondary blocking maximum in SingleMtn is found towards the western end of the high-pressure anomaly, near
the storm track exit (Fig. 6f). A tertiary, but relatively weak blocking maximum is found from roughly 150° W – 110° W,
where U250 contains a local minimum in between the two U250 maxima. The blocking in this region is a probable explanation
for the gap in the U250 maximum, as blocks are known to inhibit or even halt zonal flow. The second and third blocking
maxima are consistent with current theory linking blocking to Rossby wave-breaking (Pelly and Hoskins, 2003; Berrisford et
al., 2007; Masato et al. 2012), which as mentioned before, predominantly occurs at the storm track exit. Each of the 3 blocking
maxima in SingleMtn are found to be unique regions of block genesis.
The presence of a second, symmetrically placed mountain in SymMtn leads to the occurrence of significantly more
blocking than in the aquaplanet, SingleMtn, and even AsymMtn (Fig. 6g and Table 2). The blocking maxima in SymMtn sit
near the intersection of the high-pressure anomaly and storm track exit (Fig 6c and Fig. 6g). In AsymMtn there are blocking
maxima also on the windward sides of the mountains near each respective high-pressure anomaly (Fig. 6h), and the overall
area-averaged block frequency is slightly greater than SingleMtn, but less than SymMtn (Table 2).
In AsymMtn Wide Basin, the blocking maximum is close to the stationary wave maximum and a secondary blocking
maximum occurs at the western edge of the high-pressure anomaly, near the storm track exit (Fig. 6d and 6h). As in SingleMtn,
these separate maxima correspond to distinct block genesis regions.
In AsymMtn, the short basin has a greater block frequency maximum than wide basin (Fig. 6h). Like SymMtn, the
short basin in AsymMtn has a storm track exit region that overlaps with the high-pressure maximum of the stationary wave
when compared to SingleMtn and AsymMtn Wide Basin. This perhaps explains the enhanced blocking climatological
maximum in AsymMtn Short Basin compared to AsymMtn Wide Basin. On the other hand, AsymMtn Short Basin has such a
small zonal extent that the storm track exit overlaps with the mountain. Thus, in this short basin there are more likely to be
times in which storm development occurs just upstream of the mountain – and such conditions would inhibit blocking.
As observed in the Atlantic basin of Earth, we suspect the shortened jet in AsymMtn Short Basin acts as a waveguide
that funnels transient eddies and $\vec{W}$ into the anticyclonic anomaly of the stationary wave, and these eddies have the potential
to feed blocks or help destroy them. The details of those processes are a focus of future work.
When comparing the blocking climatologies for each configuration, we find that blocks are predominantly generated
at high-pressure stationary maxima, regions dominated by wave breaking (storm track exit), or at some spatial mixture of the
two (Figs. 6e-6h). The aquaplanet shows that blocks can arise purely from eddy-eddy interactions, whereas the other
configurations show that blocks can also be induced by topography, at a more frequent rate.
We want to highlight the result that SymMtn has the largest area-averaged block frequency (Table 2) and number of
events (Table 3) out of all the configurations. We hypothesize that this is because the ocean basins in SymMtn have a zonal
extent that allows a synergy between the block genesis mechanisms associated with the high-pressure anomaly induced by the



topography and block maintenance mechanisms associated with the storm track exit. A similar inference can be made when
comparing the short and wide basins of AsymMtn, where the short basin contains a stronger blocking maximum and a more
spatially coincident storm track exit with the high-pressure of the stationary wave. However, as mentioned above, AsymMtn
Short Basin is so short that there is not enough spatial separation between the storm track entrance and the downstream
topographical feature. The processes governing the interactions of the storm tracks and the topographical features in relation
to blocking are topics of future work.

**3.4 Block duration and displacement**

One of the characteristics that allows blocks to influence midlatitude weather is their persistence. As such, we examine

the influence of topography on block persistence using our duration metric. First, we find that adding topography, regardless
of configuration, leads to longer duration blocks on average (Table 3). For SingleMtn, the difference compared to Aquaplanet
is statistically significant (8.4 versus 7.3 days, respectively). For SymMtn and AsymMtn, the mean duration is longer, but the
difference is not significant at the 85th percentile. This is because of the large variance in block duration when we consider all
midlatitude blocks generated by the model. However, if we subset the blocks, based on the location in which they are generated,
this result changes.

Using the regions defined in Table 1, we found that for blocks that occur in the eastern portion of the ocean basins

(i.e., East Blocks), the mean durations are all significantly longer than those of aquaplanet (Table 4, and see Fig. 8.b. for the
probability density distributions). The east portion of the ocean basins is near the local maxima in the stationary wave – west
of the topography, thus, the longest duration blocks per configuration are those that are generated just upstream of topography.
Furthermore, the average duration values for the East blocks in SingleMtn and AsymMtn are greater than their Mid
counterparts (i.e. blocks that start near the storm track exits, see Table 1 and Fig. 6). SingleMtn East and SingleMtn Mid have
mean block durations of 9.1 and 8.2 days respectively (Table 4). AsymMtn Wide Basin East and AsymMtn Wide Basin Mid
have significantly different mean block durations of 8.3 and 7.1 days respectively (Table 4). Thus, blocks that form near
topography in this model (i.e., the East blocks), tend to persist for longer times than those that form far from topography (i.e.,
Mid blocks, or blocks in the aquaplanet configuration). The same analysis applied to the NH and SH in reanalysis found that
the NH, which presumably contains much more topographically forced blocks, has an average block duration (8.0 days) that
is significantly longer than those from the SH (6.9 days).

A natural question to ask then is: Why do blocks originating near topography have longer durations than those in the

aquaplanet? Given that the topographic configurations contain both stronger localized storm track regions and blocks generated
by topography, whereas the aquaplanet does not, we hypothesize two possible explanations for the differences in duration:
1.   The stronger localized storm tracks create more eddies, which would provide more transient eddies that could

feed the blocks through dry dynamics (e.g., Shutts, 1983; TN01; Yamazaki and Itoh, 2013) or moist dynamics

(Pfahl et al., 2015).



2.   Topographically generated blocks last longer than blocks predominantly generated by eddy-eddy interactions
because they are fundamentally different.

Regarding Hypothesis 2, we analyzed $\vec{W}$ composites for East Blocks as compared to Mid Blocks and found minimal
differences aside from increased composite $\vec{W}$ magnitudes for the Mid Blocks (Fig. 9). Since the Mid blocks are those more
likely to be generated by eddy-eddy interactions, this result refutes Hypothesis 2. Related to this, as discussed in Sect. 3.1, the
life cycle composites of the wave activity flux are very similar for the aquaplanet configuration and the configurations with
topography (i.e. Fig 3 and Fig. 9). These results point more toward hypothesis 1 but are very much preliminary. More work is
planned to investigate the maintenance of blocking between the model integrations.

Next, we test for differences in block displacement to determine if topography obstructs the movement of blocking
events. The differences in average block displacement (Table 3) between the four configurations are small. When comparing
East, Mid, and the aquaplanet blocks, the differences are also small. Even when isolating just the AsymMtn Wide Basin East
and AsymMtn Short Basin blocks, the difference in average block displacement is still slight. Thus, even with topographic
obstructions, block displacement is not affected.

**4. Summary and conclusions**

This work utilizes an idealized moist GCM to better understand atmospheric blocking. We start with an analysis of
blocking in an aquaplanet. Then we systematically add topographic features to investigate the influence of topography on
blocking, in terms of their climatological location, duration, and displacement.

In the aquaplanet we find that blocks can be generated purely through eddy-eddy interactions; i.e., they do not require
surface forcing. This result agrees with Hu et al. (2008) and Hassanzadeh et al. (2014), which are, to our knowledge, the only
other aquaplanet studies related to blocking. To expand on the results of those previous studies, we qualitatively examined the
dynamical life cycle of the blocks in the aquaplanet. Block centered composites of Z500' and $\vec{W}$ show that block lifecycles
include large-scale Rossby wave features with $\vec{W}$ entering the block during onset, followed by concentrated $\vec{W}$ inside the block
during peak strength, and ending with $\vec{W}$ emitted downstream of the block into low-pressure regions during decay. This
behavior is similar to what is found in nature (e.g., TN01).

Like Berrisford et al. (2007), who looked at blocking in Earth's Southern Hemisphere, we identify distinct high-
latitude blocking events that differ from midlatitude blocks. High-latitude blocks in the aquaplanet have lower geopotential
height anomalies, primarily occur poleward of the main zonal channels of $\vec{W}$, and do not contain strong concentrations of $\vec{W}$
at peak strength. This suggests an alternative maintenance mechanism for high-latitude blocks than those proposed for blocks
in general by Nakamura et al. (1997) and TN01. High-latitude blocks are also identified in the topographic configurations, but
to a lesser extent than the aquaplanet.



For the topography experiments, we modified the aquaplanet model in the following ways: (1) adding a single 3-km
mountain; (2) adding two 3-km mountains evenly spaced with respect to longitude; and, (3) adding two 3-km mountains
asymmetrically space with respect to longitude, to create one long and one short ocean basin.
The addition of topography led to some changes in blocking that were universal across all configurations with
topography, compared to the aquaplanet integration:
-    There are localized maxima in blocking, upstream of topography; near the high-pressure maximum of the stationary
waves; a source region of anticyclonic vorticity.

-    There is significantly more wintertime blocking overall with topography present.
-    When topography is present, blocks have longer durations.
-    Topography does not play a key role in determining the characteristics of block movement.
Based on ERA-Interim reanalysis, these results mirror what is observed for the NH and SH, where the NH contains more
topography, blocking, and longer lasting blocks.
The addition of topography also induces stationary waves, and localized maxima in the jet streams and the storm
tracks. This response has been documented previously, but our interest was the interaction between these features and blocking.
In all configurations with topography, local blocking maxima are found near high-pressure stationary anomalies as well as
storm-track exit regions, where Rossby waves tend to break. A local minimum in blocking is coincident with the jet stream
maxima and storm track entrance regions.
The spacing between the two mountains is important for the amount of blocking that is produced: symmetrically
placed mountains leads to significantly more blocking than all other configurations. Both blocking maxima in SymMtn
spatially overlap with their ocean basin's respective storm track exit region and anticyclonic stationary anomaly. We suspect
SymMtn's increased block frequency reflects a spatial resonance between breaking Rossby waves at the storm track exits
interacting with high-pressure anomalies generated by the mountains. This helps explain some of the differences in the
blocking climatology we observe between the Pacific and Atlantic in the NH.
Though the blocking maxima in the NH Atlantic and Pacific basins are similar in magnitude, the Pacific maximum
covers a larger area – thus there is more blocking in the Pacific. In the NH Pacific, a similar spatial distribution to SymMtn is
observed between the storm track exit, blocking maximum and stationary wave induced by orography. The Atlantic on the
other hand, is akin to the Short Basin in AsymMtn, a storm track whose exit and maximum both are semi-coincident with the
stationary high-pressure and blocking maximum. Our results suggest the broader Pacific blocking maximum is a consequence
of better spatial resonance between the Pacific storm-track exit and stationary anomaly, compared to the Atlantic. An
alternative hypothesis is that the semi-coincidence between the storm track and blocking maxima in the Atlantic inhibits
blocking. Another possible explanation is that the stationary wave in the Atlantic is forced by land/sea contrasts rather than a
mountain, leading to different interactions with its storm track, as compared to the Pacific. Further work will be done to
investigate the sensitivity of climatological blocking maxima to the location of storm track exits.





In the configurations with topography, blocks generated near topography last longer, on average, than those produced
away from topography. However, compositing results of Z500' and $\vec{W}$ found blocks forming near and away from topography
yielded little differences aside from blocks away from topography interacting with larger magnitudes of $\vec{W}$ compared to near
topography counterparts. Further work is planned to provide a mechanistic explanation for these differences we find in block
duration.
Overall, this work elucidates fundamental information on the formation, dynamical evolution, spatial distribution,
duration, and displacement of atmospheric blocking. Future work will utilize a suite of dynamical diagnostics to take a deeper
look into the differences between blocks generated near topography compared to those that are not, and how it relates to what
is observed in reality.






**Acknowledgements:**
This study is supported and monitored by The National Oceanic and Atmospheric Administration – Cooperative Science
Center for Earth System Sciences and Remote Sensing Technologies under the Cooperative Agreement Grant #:
NA16SEC4810008. The authors would like to thank The City College of New York, NOAA Center for Earth System Sciences
and Remote Sensing Technologies, and NOAA Office of Education, Educational Partnership Program for fellowship support
for Veeshan Narinesingh, and the American Society for Engineering Education for their support of Spencer K. Clark through
a National Defense Science and Engineering Graduate Fellowship. The statements contained within the manuscript are not the
opinions of the funding agency or the U.S. government, but reflect the authors' opinions.





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



| Configuration - Region | Western Edge | Eastern Edge |
|---|---|---|
| SingleMtn - East | 10˚ W | 90˚ E |
| SingleMtn - Mid | 160˚ W | 60˚ W |
| SymMtn - East | 10˚ W | 90˚ E |
| AsymMtn – Wide Basin East | 10˚ W | 90˚ E |
| AsymMtn – Wide Basin Mid | 110˚ W | 10˚ W |
| AsymMtn - Short Basin | 110˚ E | 150˚ W |

**Table 1: Regions used for subsetting blocks in certain analysis, per model configuration. Each region spans 30˚- 65˚ N, for the 100˚**
**of longitude listed in the table.**





| Configuration | Area Averaged Block Frequency (%), 30° N- 65° N | Area Averaged Block Frequency (%), 65° N- 90° N | Area Averaged Block Frequency (%), 30° N- 90° N |
|---|---|---|---|
| Aquaplanet | 1.98 | 1.69 | 1.93 |
| SingleMtn | 2.53 | 1.46 | 2.34 |
| SymMtn | 3.01 | 1.35 | 2.71 |
| AsymMtn | 2.58 | 1.35 | 2.36 |

**Table 2: Area-averaged, wintertime block occurrence frequency for midlatitudes and high-latitudes in all idealized model configurations.**






| Configuration | Number of Events | Mean Block Duration (days) | Mean Block Displacement per 6 hours (km) |
|---|---|---|---|
| Aquaplanet | 95 | 7.3 | 155.3 |
| SingleMtn | 105 | 8.4 | 152.6 |
| SymMtn | 139 | 8.0 | 150.3 |
| AsymMtn | 125 | 7.6 | 158.2 |

**Table 3: Total count of blocking events mean block duration, and mean block displacement, for midlatitude winter blocks in each**
**configuration.**






| Configuration - Region | Number of Events | Mean Block Duration (days) |
|---|---|---|
| Aquaplanet – All longitudes | 95 | 7.3 |
| SingleMtn - East | 57 | 9.1 |
| SingleMtn - Mid | 27 | 8.2 |
| SymMtn - East | 51 | 8.8 |
| AsymMtn - Wide Basin East | 42 | 8.3 |
| AsymMtn - Wide Basin Mid | 31 | 7.1 |
| AsymMtn - Short Basin | 50 | 7.2 |

**Table 4: Average midlatitude winter block duration and number of events for blocks sorted by configuration and select basins as**
**defined in Table 1.**





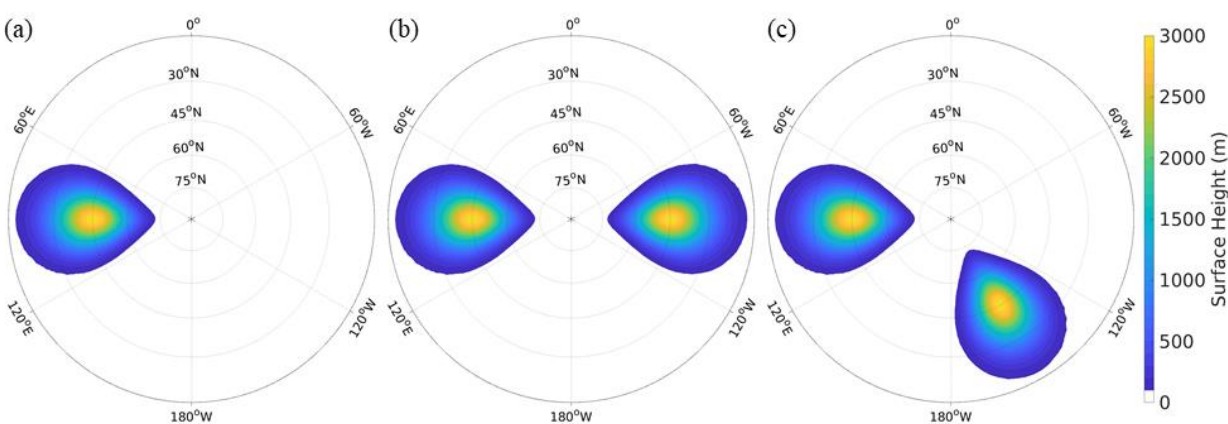


**Figure 1: Surface heights (shading) of the 3 topographical configurations of the idealized model: (a) SingleMtn (b) SymMtn (c) AsymMtn.**






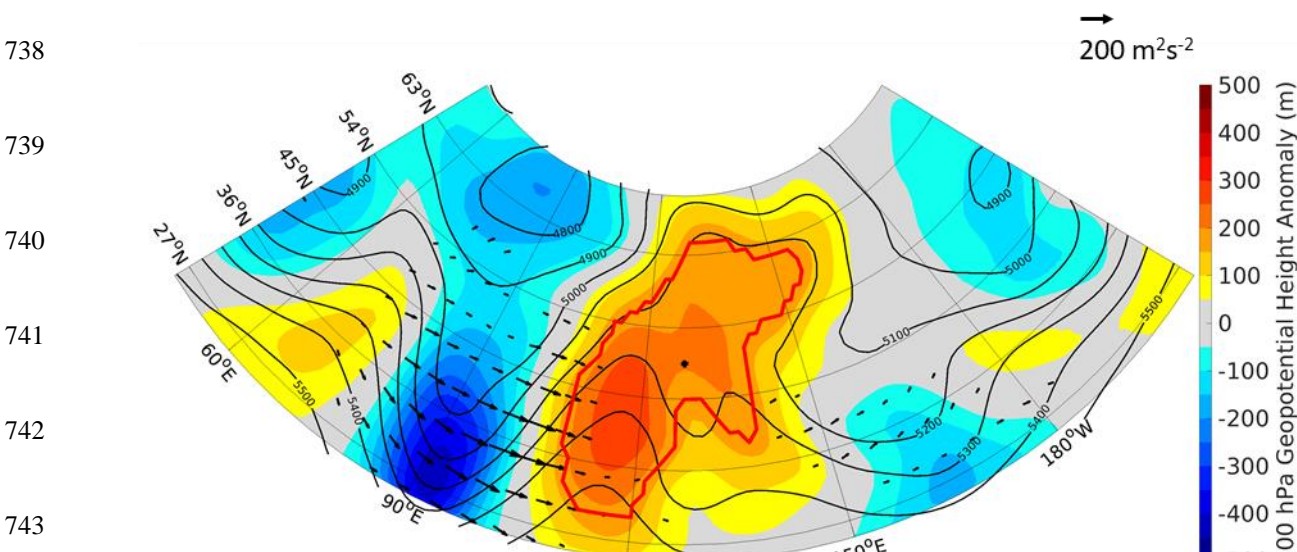

**Figure 2: 500 hPa geopotential height (black contours), 500 hPa geopotential height anomaly (shading), outline of blocked area (red contour), and wave activity flux vectors, $\vec{W}$ (black arrows), for the first day of a blocking episode in the aquaplanet run. The black dot inside the block denotes the block centroid. Geopotential height contours are in 100 m intervals. Only $\vec{W}$ with magnitudes greater than 25 m$^2$s$^{-2}$ are shown.**



751

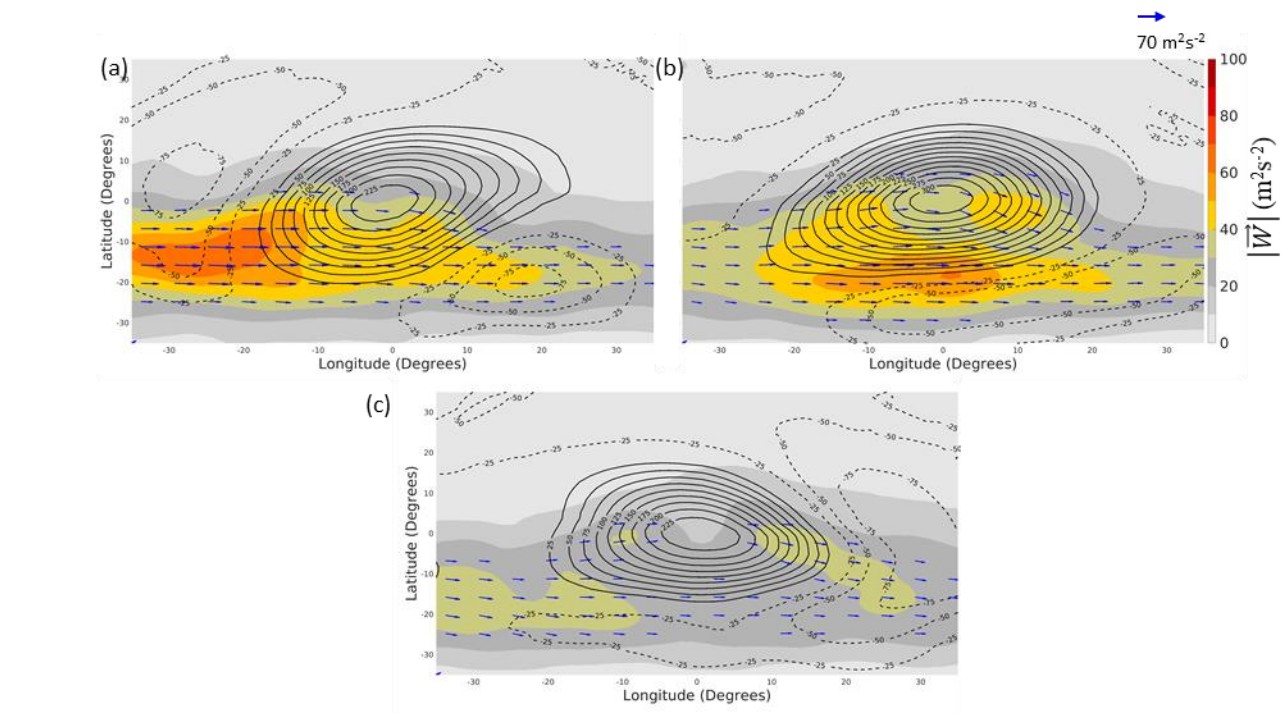

767

**Figure 3: Block centered composites of positive 500 hPa geopotential height anomalies (solid contours), negative 500 hPa geopotential height anomalies (dotted contours), $\vec{W}$ (blue arrows), and $\left|\vec{W}\right|$ (shading) for midlatitude blocks in the aquaplanet. (a), (b), and (c) are composites over the first, strongest, and last timesteps of blocking episodes, respectively. 500 hPa geopotential height anomaly contours are in 25 m intervals. $\vec{W}$ with magnitudes less than 25 m² s⁻² are removed.**

772

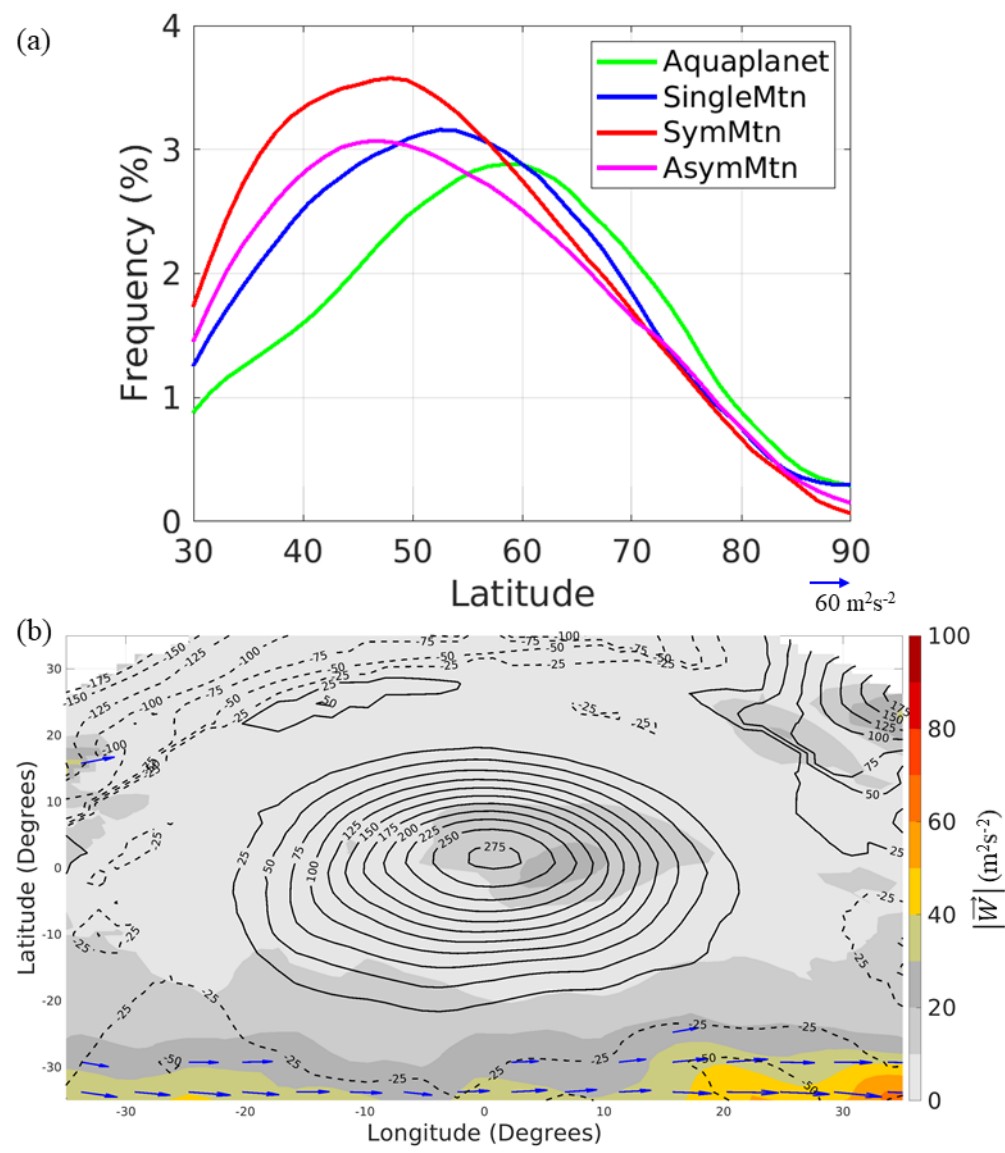

773

774

**Figure 4: (a) Zonally averaged winter blocking climatology for each model configuration (b) For high-latitude aquaplanet blocks during peak intensity, block centered composites of positive 500 hPa geopotential height anomalies (solid contours), negative 500 hPa geopotential height anomalies (dotted contours), $\vec{W}$ (arrows), and $\left|\vec{W}\right|$ (shading). 500 hPa geopotential height anomaly contours are in 25 m intervals. $\vec{W}$ with magnitudes less than 25 m$^2$ s$^{-2}$ are removed.**









**Figure 5: (a-b) Left: Winter stationary wave (shading) and U250 climatology (heavy black contours) for the (a) northern and (b) southern hemispheres. U250 contours are in 10 m/s intervals. (c-d) Right: Winter blocking climatology (shading) and storm-track (heavy black contours) for the (c) northern and (d) southern hemispheres. Storm track contours are in 4 m intervals.**








**Figure 6: (a-d) Left: Winter stationary wave (shading) and U250 winter climatology (contours) for the (a) aquaplanet (b) SingleMtn (c) SymMtn (d) and AsymMtn. U250 contours are in 10 m/s intervals. (e-h) Right: Winter blocking climatology (shading) and storm-track (contours). Storm-track contours are in 2 m intervals. Pink and black dotted contours represent surface height, where the outer contour is the edge of the land-mask, and the inner contours represent 1, 2, 3 km respectively. Results are presented for (e) aquaplanet (f) SingleMtn (g) SymMtn (h) AsymMtn. The red outlines in the e-h indicate the regions used when separating blocks by region.**



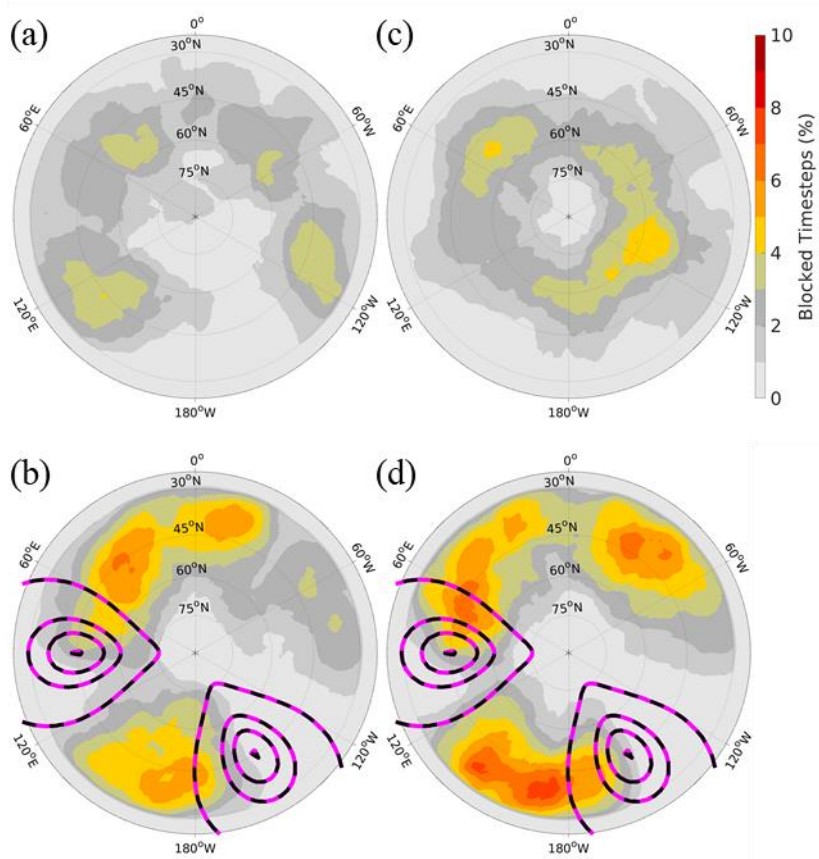

**Figure 7: Winter blocking climatologies (shading) computed by randomly sampling (a, c) 15 years of years 11-40 in the aquaplanet, and (b, d) 15 years of years 11-40 in AsymMtn. Pink and black dotted contours represent surface height, where the outer contour is the edge of the land-mask, and the inner contours represent 1, 2, 3 km respectively.**



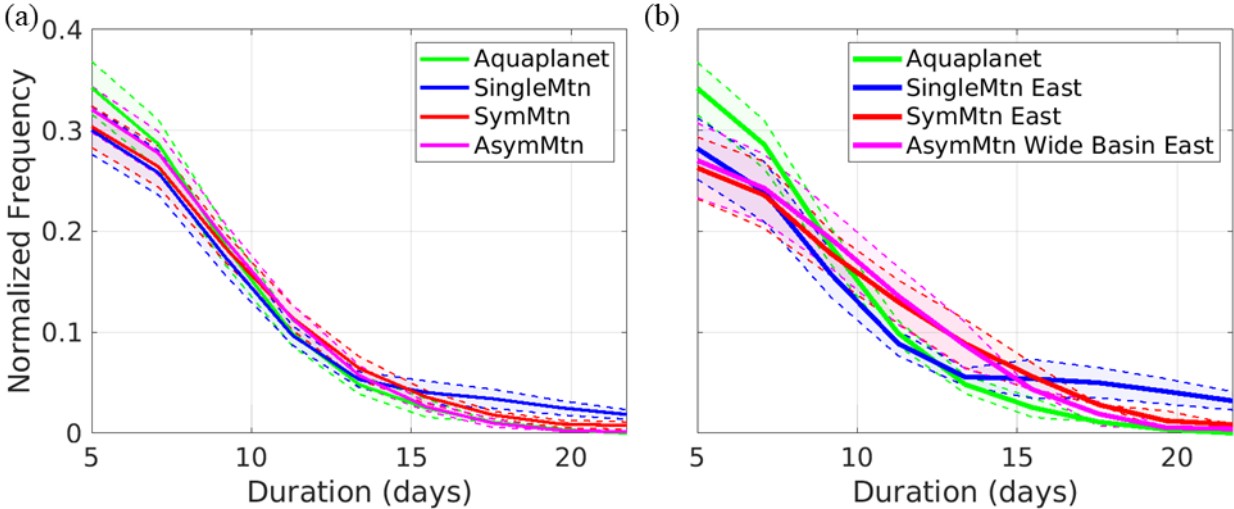

801

**Figure 8: Normalized Block Duration Probability Density Distributions for (a) all winter blocks within each model configuration, and (b) Aquaplanet and just the East blocks for each topographic configuration. Thick colored lines denote the mean probability density distribution for each configuration. Shaded regions bordered by dotted lines outline +/- half a standard deviation from the mean.**

806

807



**Figure 9: Block centered composites of positive 500 hPa geopotential height anomalies (solid contours), negative 500 hPa geopotential height anomalies (dotted contours), $\vec{W}$ (blue arrows), and $\left|\vec{W}\right|$ (shading) for (a-c) Left: SingleMtn midlatitude Mid and (d-f) Right: SingleMtn midlatitude East blocks. The top panel (a, d), middle panel (b, e), and bottom panel (c, f) are composites over the first, strongest, and last timesteps of blocking episodes, respectively. 500 hPa geopotential height anomaly contours are in 25 m intervals. $\vec{W}$ with magnitudes less than 25 m$^2$ s$^{-2}$ are removed.**