# Peer review of "Atmospheric Blocking in an Aquaplanet and the Impact of Orography"

_Weather and Climate Dynamics, 2020_

## Referee Comment (RC1) · Anonymous Referee #1 · 13 Feb 2020

General comment

This paper studies the topographic effect on blocking formation, using an idealized GCM. The authors have done aquaplanet simulations and simulations with different types of topographies (idealized mountains). They compared the simulation results with and without topographic forcing to demonstrate the influences of topography on blocking formation in terms of dynamics, spatial frequency, duration and displacement. They conclude that the simulation results have important implication for understanding blocking dynamics in the real atmosphere. Overall, the paper is interesting and clearly written, and it would certainly improve our understanding in blocking dynamics, which suddenly became a hot research topic in recent few year. I would recommend publication with minor revision. My comments in the following are for the authors' reference.

[Figure]

Specific comments

1. In the simulation by Hu et al. (2008), solar insolation is fixed at March equinoctial condition. It generates greater meridional temperature gradients in the middle and upper troposphere and thus stronger baroclinic eddies. This could be the reason why there are frequent blocking events in their simulations. In the present study, if insolation has seasonal variations, it would be good to look at whether there are seasonal variations of blocking frequencies.

2. It would be good if the authors add a couple of sentences about why the mountain size of 15 degrees in latitude and longitude is chosen. Is it large enough to generate stationary waves?

3. Line 247: Why do you choose the 85% confidence level? Is it too low? People usually use at least the 90% confidence level.

4. Line 51: may be key –> may be the key

5. Line 108: Q-flux, is there horizontal heat flux?

6. Section 2.4: there are too many short paragraphs. It would be good to put them together.

7. Line 505: surface forcing –> topographic forcing 8. Lines 537 and 545: "resonance" may not be a good terminology. It is actually nonlinear eddy-eddy interaction or interaction between transient waves and stationary waves.

9. Fig. 4b: I am confused by this plot at beginning, and I thought blockings occur at the equator. It is good to pointed out in the figure caption that the reference latitude is removed.

---

## Referee Comment (RC2) · Anonymous Referee #2 · 13 Feb 2020

**REVIEW: Atmospheric blocking: The impact of topography in an idealised general circulation model by Narinesingh et al.**

**Summary:**

Scientific significance: Fair
Scientific quality: Fair
Presentation quality: Fair

This paper uses an idealized aquaplanet model to compare statistics of atmospheric blocking between configurations with zonally symmetric and asymmetric surface boundary conditions. Zonally asymmetric boundary conditions change the spatial location, frequency and duration of blocking in comparison to the zonally symmetric configuration, consistent with changes in climatological storm tracks and stationary waves. The results suggest zonally asymmetric surface boundary conditions control the spatial distribution of blocking in the real atmosphere to first order.

I think this paper is interesting and the results are relevant to this journal. However, I think the paper 1) does not provide sufficient explanations for the questions posed, 2) needs to focus more on the key results and 3) does not consider a key implication of the experiments which was proposed in previous work. Therefore it is for these reasons, which are summarised in more detail below, which I recommend major revisions before this paper can be published.

**General comments:**

1. I don't think the paper provides sufficient explanations for the questions posed (e.g. lines 344-346 and 483-487). Specifically, the explanations are generally qualitative and show consistency between different fields (e.g. storm tracks, stationary waves and blocking) and the authors often state that future work is required to understand the causal mechanisms (e.g. lines 443-445, 457-458, 492-493, 556-558). While it is clear that the surface boundary conditions cause the changes in blocking, it is difficult to establish the exact mechanisms because everything is changing at once. Therefore, I'm not sure the authors can answer the questions posed with these simulations only. It likely requires more detailed analysis with regards to the theories discussed in the Introduction or more experiments with simpler models.

2. I think the paper would benefit from focusing more on the key results. For example, I'm not sure how the analysis of high-latitude versus low-latitude blocking relates to the experiments because the authors state that the results are similar in all simulations and reanalysis (lines 296-299) and blocking is much less frequent in high-latitudes (Fig. 4a). The authors devote a significant portion of the results to discussing the reanalysis and model climatological stationary waves, storm tracks and jets (lines 306-404) which could be summarised in a few sentences since these features are well known and the responses are well understood. Finally, the subsampling analysis in Fig. 7 could also be discussed in words only and the case study in Fig. 1 could be omitted altogether since similar results are presented in Fig. 2.

3. I think the paper does not consider a key implication of their results which was proposed by

Hu et al. (2008). Viewed from their perspective, the results presented here demonstrate that zonally symmetric models capture the key features of blocking. To be clear, the results show that the surface boundary condition controls the spatial distribution of blocking. However, I was surprised to see that many of the hemispheric statistics listed in Tables 2-4 show modest changes on the order of 10-30% when topography is included. Moreover, the composite analyses in Figs. 3 and 9 suggest the dynamics of individual blocks are similar with and without topography. I think this would be an interesting point given recent work has focused on the role of orographic drag in improving the simulation of blocking (Pithan et al. 2016 GRL) and zonally asymmetric boundary conditions have been hypothesised to be critical for blocking formation (e.g., Tung and Lindzen 1979). Moreover, the results suggest that the poor simulation of blocking in climate models for the past several decades (e.g., Davini and D'Andrea 2016 JCLIM) could be better understood by understanding blocking dynamics in more simple aquaplanet models.

Given this different perspective and the issues discussed in general comment 1, a suggestion to improve the paper would be to focus on the following questions: 1) Are the characteristics of individual blocking events different with zonally symmetric versus asymmetric boundary conditions? 2) do zonally asymmetric boundary conditions control the spatial statistics of blocking? and 3) Are the hemispherically integrated statistics of blocking different for zonally symmetric versus asymmetric boundary conditions ?

**Specific comments:**

1. Lines 18-19: This suggests high-latitude blocking is different from reanalysis in the model however the text says the opposite.

2. Lines 42-43: Is this true if you integrate blocking statistics over the entire NH versus SH? How different are the statistics quantitatively?

3. Line 46: I think a better topic sentence for this paragraph is that the dynamics of blocking are unclear. Also I suggest to cite Nakamura et al. (2018) Science. Their work provides a simple theory for which can be used to explain why stationary waves preferentially localise blocking in certain longitudes, e.g., they slow the 'speed limit' and modify the source of zonal wave activity flux.

4. Lines 72-74: Suggest adding 'in order to relate the idealized results to the real atmosphere, e.g. NH vs SH and NH PAC vs NH ATL'.

5. Line 94: Does the omission of these processes influence blocking in the model compared to the real world ? e.g. diabatic effects shown by Pfahl et al. (2015) nature.

6. Line 96: The experiments include both topography and land-sea contrast yet the title only mentioned topography. What is more important for the results, topography or land-sea contrast?

7. Line 99: Suggest mentioning again why this specific configuration is used: to relate results to the real atmosphere.

8. Lines 100-106: Have the authors confirmed how their results are sensitive to the mountain amplitude?

9. Section 2.3: Could the anomaly normalisation or the spatial area threshold used to identify events be responsible for the different blocking events in mid versus high latitudes? Longitude lines converge poleward and the thresholds were likely tuned for midlatitudes. Have the authors checked the sensitivity of their results do different thresholds? Or a different blocking index? I suggest confirming the results with a simpler index involving only geopotential height anomalies or the reversal of the geopotential.

10. Sections 2.4.1-2.4.2: I suggest mentioning this in words in the results instead.

11. Section 2.4.3: Isn't a simple lanczos filter more commonly used (e.g. Shaw et al. 2016 nature)?

12. Section 2.4.5: I'm confused about the wave activity flux vectors. Shouldn't these be calculated for high-frequency eddies only since they characterize their influence on low-frequency blocking ? e.g., Hoskins et al. 1983 JAS Fig. 15. Here the quantities used to calculate the fluxes are low pass filtered.

13. Lines 247-248: I suspect that the lower statistical significance threshold was used because the blocking statistics are not that different between the zonally symmetric versus asymmetric experiments. This supports general comment 3 above.

14. Lines 290-291: I disagree. The contours differ by 25m, e.g. 275 versus 300.

15. Lines 505-506: I believe Hassanzadeh et al. 2014 used a dry-dynamical core not an aquaplanet model.

16. Lines 537 and 545: Resonance has a very specific meaning, e.g., multiple reflection of waves on turning points following linear theory. I don't think it is what is implied here.

17. Figs 2,3,4 and 9 and related analysis: I suggest the authors interpret the wave activity fluxes with regards to flux convergence not the flux itself since this is the key dynamical quantity for blocking (Hoskins et al. 1983 JAS, Nakamura et al. 2018 science).

18. Figs. 3 and related analysis: I suggest the authors compare the zonally-symmetric and asymmetric model simulations with reanalysis explicitly rather than reference previous work. Specifically, I suggest replacing Fig. 3 with a 3 x 3 panelled figure showing midlatitude blocking for reanalysis (top), zonally symmetric model (middle) and one zonally asymmetric model simulation for all 3 lifecycle stages (left, middle, right). This would also show that the two model configuration show similar results.

---

## Referee Comment (RC3) · Anonymous Referee #3 · 20 Feb 2020

The authors have used an idealized moist GCM and investigated some of the spatial and temporal characteristics of blocking events in the absence and in the presence of topography. I find the objectives of the paper and its results interesting and important (although further clarifications are needed). The paper is well structured and well written. I have a number of major and minor comments, which are listed below.

Recommendation: major revision

Major comments My major concern is that the paper is focused on too many questions, which have made the answers sometimes a bit too speculative. It appears to me that the three main questions are

1- Do the blocking events in aquaplanet simulations have the same dynamics as those

of the real blocking events? This is a great question and its answer has important implications for our understanding of the dynamics of the blocking events, as for example, some blocking theories require zonal asymmetries in boundary conditions/forcings. The studies of Hu et al. (2008 GRL), Hassanzadeh et al. (2014 GRL), and more recently Nabizadeh et al. (2019 GRL) have shown the existence of blocking events in aquaplanet simulations and report some of their characteristics, but certainly, there is a need for further investigation, and I am glad that these authors have focused on this question. Given the importance of the answer, I believe that the statement in Lines 296-298 needs more support. To start, I suggest that you show the analysis of Fig. 3 for the ERA data as well, so that the readers can see the comparison side by side (rather than being referred to other papers such as TN01).

2- Do the high-latitude blocks have the same dynamics as those of the midlatitude blocking events? The discussion in lines 286-292 is too speculative. I suggest that you show the analysis of Fig. 3 but for high latitude blocks (rather than the single panel in Fig. 4). Regarding the difference in dynamics: given the lack of W and weakness of the anomalies (pointed out in lines 290-291), is it possible that the high-latitude blocks are just cut-off highs that appear stationary because the zonal wind in the high latitudes is weak? (so there is really no maintenance mechanism?) What is the time scale of zonal advection in the high latitudes of the models (and what is it in the midlatitudes?)

3- What is the effect of topography on the duration, distribution, and dynamics? I think here the most interesting analysis is the comparison between Fig. 3 and 9. Whether the life cycle and dynamics are affected by the topography or not is an important question, but is barely explored. I suggest that you further elaborate on these results. Otherwise, given the very idealized nature of topography here, I am not sure how much we can learn from the distribution and duration of different simulations with different topography configurations.

Minor comments

L186: W is given in . . ...

L247: 85% is too low. I suggest using a 95% confidence interval.

---

## Author Comment (AC1) · 8 May 2020

**Response to reviewers**

**General Comments**

We are very grateful for each reviewer's input and have adjusted the manuscript accordingly. Outlined below are the major changes we have made, followed by specific responses to each reviewer. Thank you to everyone involved.

**Major Changes:**

- Further emphasize the aquaplanet's ability to generate realistic blocks, comparing them to results from reanalysis and idealized model integrations with orography (Section 3.1 of the revised manuscript)

- Removal of the midlatitude vs. high latitude blocking subsection

- Switched the orographic configurations that are analyzed to instead be single mountain configurations of varying height, and just one two-mountain configuration.

- Removal of the analysis on block displacement

- Refocusing of the questions being addressed of this paper (as per the suggestions of reviewer 2):

  1. Are blocks in an aquaplanet dynamically similar to blocks in orographically forced simulations and reanalysis?

  2. Does the presence of orography affect the overall frequency of blocking?

  3. How does orography affect the spatial distribution of blocking frequency?

  4. Does orography affect the duration of blocking events?

Reviewer feedback led us to a better appreciation of the aquaplanet results (i.e. Reviewer 2 – General Comment 3, Reviewer 3 – Major Comments 1 and 3). Therefore, we have further emphasized this section and included results that compare how blocking in an aquaplanet relates to blocking in the real-world and idealized model configurations with topography (Fig. 3).

With regards to the midlatitude vs high-latitude blocking results from the original submission, we received mixed feedback from the reviewers (i.e. Reviewer 2 -- General Comment 2, Reviewer 3 Major Comment -- 2). We acknowledge the dissimilarities between blocks in the midlatitude and high-latitude blocks, especially with regards to wave activity flux, however, we do not want to distract from the primary focuses of this paper. We have opted to remove this analysis.

Motivated by Reviewer 2 – General Comment 1, to mitigate difficulties from the interference of forcing from multiple mountains, we now choose to analyze a different set of idealized model configurations with topography. For this, the results from single mountain

configurations of varying height are presented as the new primary focus. Results from the original two-mountain configuration with zonally asymmetric spacing between the mountains are also briefly presented to reaffirm results from the single mountain analysis. Overall, the main points remain the same. Topography leads to:

1. An overall hemispheric increase in blocking frequency
2. The anchoring of regions of enhanced and suppressed blocking frequency
3. The suggestion of regions of enhanced blocking duration

To minimize redundancy, the results from the configuration with zonally symmetric spacing between the mountains from the original submission is now omitted. The discussion of the spacing of mountains having an effect on the spatial distribution of block frequency has also been removed and will be explored in future work.

The analysis on block displacement was motivated by the changing length of the ocean basins used in the previous iteration of this paper. This ended up being a very short section that produced a null result. The switch in topographic configurations now used in the revised article offers little relevance to the displacement analysis. Furthermore, we do not want to distract the reader from the overall main points of this article. The block displacement analysis is now removed.

**Response to Reviewer 1**

*General comment:*
*This paper studies the topographic effect on blocking formation, using an idealized GCM. The authors have done aquaplanet simulations and simulations with different types of topographies (idealized mountains). They compared the simulation results with and without topographic forcing to demonstrate the influences of topography on blocking formation in terms of dynamics, spatial frequency, duration and displacement. They conclude that the simulation results have important implication for understanding blocking dynamics in the real atmosphere. Overall, the paper is interesting and clearly written, and it would certainly improve our understanding in blocking dynamics, which suddenly became a hot research topic in recent few year. I would recommend publication with minor revision. My comments in the following are for the authors' reference.*

[Ans] Thank you for your feedback, we hope to have addressed your criticisms below to satisfaction. Note, any comments regarding typographical errors are skipped over here, but integrated into the manuscript.

*Specific comments:*

1. *In the simulation by Hu et al. (2008), solar insolation is fixed at March equinoctial condition. It generates greater meridional temperature gradients in the middle and upper troposphere and thus stronger baroclinic eddies. This could be the reason why there are frequent blocking events in their simulations. In the present study, if insolation has seasonal variations, it would be good to look at whether there are seasonal variations of blocking frequencies.*

   [Ans] Compared to previous studies (Tibaldi et al. 1994, MWR; Barriopedro et al. 2010, Clim. Dynam.), we observed a similar seasonal cycle in blocking within our idealized model integrations (i.e. block frequency peaking within NH DJF or SH JJA, see figure below) but some configurations had shifts of about 1 month. To avoid ambiguity, we have chosen to change our seasonal sorting from "winter" defined as DJF, to "cool season" defined as NDJFM.

   With regards to Hu et al., they find more blocking events compared to Weidenmann et al. (2002, JoC) which uses reanalysis. This is tricky to interpret however, as Weidenmann et al. (2002, JoC) counts blocks from all seasons, including summer, which has been shown to have considerably less blocking than winter (Tibaldi et al. 1994, MWR; Barriopedro et al. 2010, Clim. Dynam.). Furthermore, Hu et al. utilizes a different block tracking algorithm from Weidenmann et al., where it has been shown that different tracking algorithms each have their own biases with respect to block frequency (Barnes et al. 2011, Clim. Dynam.). We now elaborate a bit more on this in the introduction (see lines 49-55 of the revised manuscript).

   Hu et al. speculates this increase in frequency in their model is from stronger forcing from transient eddies. We have yet to explore that in the aquaplanet used here, but the enhanced blocking frequency in the idealized model used here is consistent with

an overall weaker jet (see Figs. 4b and 5c-d in the revised manuscript), and thus enhancement of blocking (see Nakamura and Huang 2018, Science).

[Figure]

2. *It would be good if the authors add a couple of sentences about why the mountain size of 15 degrees in latitude and longitude is chosen. Is it large enough to generate stationary waves?*

[Ans] Good point, this mountain size was chosen following Lutsko and Held (2016, JAS). Our results show that it is certainly large enough to generate a considerable stationary wave (Figure 6). We hope this is clearer in lines 103-105:

"Like Cook and Held (1992), and following Lutsko and Held (2016), perturbations to the surface height are introduced in the form of Gaussian mountains centered at 45˚ N with half-widths of 15 degrees in both the latitude and longitude dimensions."

It would be interesting to investigate the atmospheric response to mountain width in the future.

3. *Line 247: Why do you choose the 85% confidence level? Is it too low? People usually use at least the 90% confidence level.*

[Ans] We received similar feedback in the other reviews. After further self-clarification we have now chosen a 95% confidence interval and more careful wording to describe quantitative differences throughout our analyses. See the methods subsection 2.5.4, lines 256-257:

"A 95% confidence interval is imposed as the significance threshold for all significance testing."

5. *Line 108: Q-flux, is there horizontal heat flux?*

[Ans] No, there is no horizontal heat flux in the oceans of the idealized model integrations used for this paper. To avoid ambiguity, we replace this line about horizontal heat fluxes with a clearer description of what no Q-flux actually means in lines 112-114:

"Ocean grid cells are represented using a slab ocean with a depth of 20 m.  For simplicity we prescribe uniformly zero Q-flux, meaning that we assume that in the time mean, the net flux of energy from the ocean to the atmosphere is zero at all surface grid cells."

6. *Section 2.4: there are too many short paragraphs. It would be good to put them together.*

[Ans] We have slightly restructured this section combining the explanation of the stationary wave and storm track into one subsection (2.4.1), and the blocking and zonal wind climatologies into another (2.4.2). These fields are grouped together in a consistent way in which they are presented in Figure 4-7, and 9.

7. *Line 505: surface forcing –> topographic forcing 8. Lines 537 and 545: "resonance" may not be a good terminology. It is actually nonlinear eddy-eddy interaction or interaction between transient waves and stationary waves.*

[Ans] Agreed, this is now removed.

8. [Ans] In original review there was no comment 8.

9. *Fig. 4b: I am confused by this plot at beginning, and I thought blockings occur at the equator. It is good to pointed out in the figure caption that the reference latitude is removed.*

[Ans] We have updated the figure caption for the revised submission (Fig. 3). We hope the line we added at the very end better clarifies things:

"Figure 3: For cool season blocking events: Block centered composites of positive 500 hPa geopotential height anomalies (solid contours), negative 500 hPa geopotential height anomalies (dotted contours), $\vec{W}$ (arrows), and $\nabla \cdot \vec{W}$ (shading). (a-c) Left: Computed with SH blocks in ERA-Interim. (d-f) Centre: Computed with blocks in the aquaplanet integration. (g-i) Right: Computed with blocks in the 3 km single mountain integration. The top, middle, and bottom rows are composites over the first, strongest, and last timesteps of blocking episodes, respectively. Positive (negative) 500 hPa geopotential height anomaly contours are in 50 m (-10 m) intervals with outer contour 50 m (-30 m). $\vec{W}$ with magnitudes less than 20 $m^2 s^{-2}$ are removed. **Latitude and longitude are defined relative to the composite block center**"

**Response to Reviewer 2**

*Summary:*
*Scientific significance: Fair*
*Scientific quality: Fair*
*Presentation quality: Fair*

*This paper uses an idealized aquaplanet model to compare statistics of atmospheric blocking between configurations with zonally symmetric and asymmetric surface boundary conditions. Zonally asymmetric boundary conditions change the spatial location, frequency, and duration of blocking in comparison to the zonally symmetric configuration, consistent with changes in climatological storm tracks and stationary waves. The results suggest zonally asymmetric surface boundary conditions control the spatial distribution of blocking in the real atmosphere to first order.*

*I think this paper is interesting and the results are relevant to this journal. However, I think the Paper:*
*1) does not provide sufficient explanations for the questions posed*
*2) needs to focus more on the key results*
*3) does not consider a key implication of the experiments which was proposed in previous work.*

*Therefore, it is for these reasons, which are summarized in more detail below, which I recommend major revisions before this paper can be published.*

[Ans] We acknowledge and find validity in these criticisms. To address them, as summarized in the cover letter, in broad terms we have:

- Reformulated our questions to address key results
- Provided greater detail in our explanations to connect our results to previous work
- Modified the selection of topographic configurations to vary topography in a way that has less degrees of freedom than the original set
- Updated the selection and presentation of the dynamical fields chosen for the results presented (i.e. presenting wave activity flux divergence instead of magnitude, presentation of climatological U250 with blocking climatology, etc.) Specific details of this are given below

*General comments:*

1. *I don't think the paper provides sufficient explanations for the questions posed (e.g. lines 344-346 and 483-487). Specifically, the explanations are generally qualitative and show consistency between different fields (e.g. storm tracks, stationary waves and blocking) and the authors often state that future work is required to understand the causal mechanisms (e.g. lines 443-445, 457-458, 492-493, 556-558). While it is clear that the surface boundary conditions cause the changes in blocking, it is difficult to establish the exact mechanisms because everything is changing at once. Therefore, I'm not sure the*

*authors can answer the questions posed with these simulations only. It likely requires more detailed analysis with regards to the theories discussed in the introduction or more experiments with simpler models.*

[Ans] This comment is addressed by the modification of the research questions and set of topographical configurations that are analyzed. Reviewer comments reflected the importance of the aquaplanet results, hence the reformulated question 1. For this, block centered compositing is utilized for the aquaplanet, topographic configurations, and reanalysis (Fig. 3, section 3.1)

   With regards to the notion of using simpler models, we have opted to present an analysis for a different set of topographic configurations. The new configurations are a set of single mountain integrations with varying max surface heights (1 km, 2 km, 3 km, 4 km) and one integration with two identical 3 km mountains. The revised manuscript also contains more explicit reference and connection to previous work (namely Nakamura and Huang 2018 Science, Nakamura et al. 1997, Takaya and Nakamura 2001)

2. *I think the paper would benefit from focusing more on the key results. For example, I'm not sure how the analysis of high-latitude versus low-latitude blocking relates to the experiments because the authors state that the results are similar in all simulations and reanalysis (lines 296-299) and blocking is much less frequent in high-latitudes (Fig. 4a). The authors devote a significant portion of the results to discussing the reanalysis and model climatological stationary waves, storm tracks and jets (lines 306-404) which could be summarized in a few sentences since these features are well known and the responses are well understood. Finally, the subsampling analysis in Fig. 7 could also be discussed in words only and the case study in Fig. 1 could be omitted altogether since similar results are presented in Fig. 2.*

[Ans] The midlatitude vs. high-latitude blocking analysis is now removed. The aquaplanet results are more focused to investigate the dynamical representation of blocking across models (Fig. 3, section 3.1)

   The section regarding model climatological responses (lines 306-404 of the original manuscript) has been made to be more concise (section 3.2.2). We still choose to keep this part to affirm our methodology and set the table for the analysis that comes after using the idealized model. We remove Fig. 7 and merge the presentation of aquaplanet convergence with Fig. 4, which is now discussed in section 3.2.1.

   Regarding your suggestion of removing figure 2, we choose to keep figure 2 to provide the reader with a quick reference to what the blocks look like on an individual basis, not just in composites. Also, figure 2 provides a snapshot of the characteristic overturning of Z500 contours (a.k.a. wave-breaking) associated with blocking, which supports the idea of this model generating realistic events.

3. *I think the paper does not consider a key implication of their results which was proposed by Hu et al. (2008). Viewed from their perspective, the results presented here demonstrate that zonally symmetric models capture the key features of blocking. To be clear, the results show that the surface boundary condition controls the spatial distribution of blocking. However, I was surprised to see that many of the hemispheric*

*statistics listed in Tables 2-4 show modest changes on the order of 10-30% when topography is included. Moreover, the composite analyses in Figs. 3 and 9 suggest the dynamics of individual blocks are similar with and without topography. I think this would be an interesting point given recent work has focused on the role of orographic drag in improving the simulation of blocking (Pithan et al. 2016 GRL) and zonally asymmetric boundary conditions have been hypothesised to be critical for blocking formation (e.g., Tung and Lindzen 1979). Moreover, the results suggest that the poor simulation of blocking in climate models for the past several decades (e.g., Davini and D'Andrea 2016 JCLIM) could be better understood by understanding blocking dynamics in more simple aquaplanet models.*

[Ans] We acknowledge the constructiveness of this comment and have made changes to the focus and set of orographic configurations for this study. For our research questions we now focus on analyzing how realistic blocks in the aquaplanet are (this result is further emphasized in the revised block centered compositing analysis (Fig. 3 section 3.1), and how the spatial distribution and duration of blocking responds to mountains.

In response to your suggestion of using simpler models, we now primarily focus on single mountain integration of varying height, rather than various configurations with multiple mountains as before. We also now cite have included the work of Pithan et al. as a reference in the discussion section, see lines 459-460:

"This configuration is like the others that include mountains in that it imposes zonally asymmetric forcing in land-sea contrast and orographic drag (Pithan et al., 2016)"

*Given this different perspective and the issues discussed in general comment 1, a suggestion to improve the paper would be to focus on the following questions:*

*1) Are the characteristics of individual blocking events different with zonally symmetric versus asymmetric boundary conditions?*

*2) do zonally asymmetric boundary conditions control the spatial statistics of blocking?*

*3) Are the hemispherically integrated statistics of blocking different for zonally symmetric versus asymmetric boundary conditions?*

[Ans] Thank you for the suggestions. We have incorporated them into the formulation of the questions being addressed in the revised version of this paper. The questions are restated below:

1. Are blocks in an aquaplanet dynamically similar to blocks in orographically forced simulations and reanalysis?
2. Does the presence of orography affect the overall frequency of blocking?
3. How does orography affect the spatial distribution of blocking frequency?
4. Does orography affect the duration of blocking events?

*Specific comments:*

1. *Lines 18-19: This suggests high-latitude blocking is different from reanalysis in the model however the text says the opposite.*

   [Ans] These lines from the abstract are removed as well as the related analysis from the manuscript.

2. *Lines 42-43: Is this true if you integrate blocking statistics over the entire NH versus SH? How different are the statistics quantitatively?*

   [Ans] Yes this holds when you integrate blocking statistics over the NH and SH. This is discussed quantitatively in results section 3.2.2, lines 350-351:

   "For the NH (SH) in this dataset, 485 (336) blocking events are found yielding a hemispherically-averaged blocking frequency of 2.7 % (1.6 %)."

3. *Line 46: I think a better topic sentence for this paragraph is that the dynamics of blocking are unclear. Also I suggest to cite Nakamura et al. (2018) Science. Their work provides a simple theory for which can be used to explain why stationary waves preferentially localise blocking in certain longitudes, e.g., they slow the 'speed limit' and modify the source of zonal wave activity flux.*

   [Ans] The original topic sentence is removed; this paragraph now begins with a discussion of the theories behind blocking explicitly. Nakamura and Huang (2018) is now more explicitly referenced through the paper, especially in regard to enhanced blocking found near the high-pressure stationary wave anomaly. The new version of this paragraph can be found in lines 64-70:

   "Previous work suggests that the spatial distribution of blocking frequency (hereafter, the blocking climatology) is dependent on the behaviour of the stationary waves, jet streams, and storm tracks. Nakamura and Huang (2018) for example, propose that blocking is most ubiquitous in regions where the positive anomaly in the stationary wave maximizes, and mean westerly flow is weak. Work by others on the effects of transient eddy forcing on blocks (Shutts, 1983; Nakamura et al., 1997; Takaya and Nakamura, 2001; Wang and Kuang, 2019), shows the importance of the storm tracks. The work presented here aims to better characterize the manner in which the spatial distribution of the stationary waves, jet streams, and storm tracks are linked to the blocking climatology."

4. *Lines 72-74: Suggest adding 'in order to relate the idealized results to the real atmosphere, e.g. NH vs SH and NH PAC vs NH ATL'.*

   [Ans] This part of the introduction has been revised to align with the overall updates.

5. *Line 94: Does the omission of these processes influence blocking in the model compared to the real world? e.g. diabatic effects shown by Pfahl et al. (2015) nature.*

[Ans] According to the work of Pfahl et al (2014)., Steinfeld et al. (2019), etc., the omission of diabatic processes certainly should have an influence on blocking. This model does include latent heat release due to the condensation of water vapor, both in the large scale and parameterized sense. The main simplification is that it does not include the impacts of clouds. See Frierson et al., 2006, JAS for more details on the model.

6. *Line 96: The experiments include both topography and land-sea contrast, yet the title only mentioned topography. What is more important for the results, topography or land-sea contrast?*

[Ans] We have updated the title to eliminate this ambiguity. We replace "topography" with "orography" to encompass changes in both land-sea contrast and lower boundary height. This is a great question, but beyond the scope of this work. With the orographic configurations used here, we cannot answer this question, however we do partially examine this topic in the discussion section where we present results from a run with a flat land patch.

7. *Line 99: Suggest mentioning again why this specific configuration is used: to relate results to the real atmosphere.*

[Ans] Explicit reminder is now included in line Section 2.2, lines 109-111:

"TwoMtn: 1 integration with two Asymmetrically placed 3 km high Gaussian mountains centered at 45° N, 90° E and 45° N, 150° W, respectively. This placement is to loosely mimic the wide (Pacific) and short (Atlantic) zonal extents of the NH ocean basins."

8. *Lines 100-106: Have the authors confirmed how their results are sensitive to the mountain amplitude?*

[Ans] This is investigated in the new set of model configurations with topography, Fig. 6, Section 3.2.3.

9. *Section 2.3: Could the anomaly normalisation or the spatial area threshold used to identify events be responsible for the different blocking events in mid versus high latitudes? Longitude lines converge poleward and the thresholds were likely tuned for midlatitudes. Have the authors checked the sensitivity of their results do different thresholds? Or a different blocking index? I suggest confirming the results with a simpler index involving only geopotential height anomalies or the reversal of the geopotential.*

[Ans] To mitigate any discrepancies related to this, we have removed the section of this paper analyzing midlatitude vs high-latitude blocking episodes. Regarding sensitivities in the blocking index, it proved impractical to implement and analyze different indices. This index however has proven be reliable, and our results are similar to that of previous work.

10. *Sections 2.4.1-2.4.2: I suggest mentioning this in words in the results instead.*

[Ans] We acknowledge this criticism but choose to maintain this structuring to provide quick, localized references of these analysis metrics for the reader. To condense things, we have combined the explanation of the stationary wave and storm track into one subsection (2.4.1), and the blocking and zonal wind climatologies into another (2.4.2). These fields are grouped together in a consistent way in which they are presented in Figure 4-7, and 9.

11. Section 2.4.3: Isn't a simple lanczos filter more commonly used (e.g. Shaw et al. 2016 nature)?

[Ans] In our experience, there are many different acceptable methods for filtering the data to isolate the transient eddies used in the calculation of the storm tracks. The Wallace et al. 1988 paper makes a point of explaining how the 24-hour differences of the daily means acts to filter the data in a similar manner to a bandpass filter (using a technique such as the Lanczos filter). The review of storm tracks by Chang et al. (2002, J. Climate) gives a brief history of the subject of time filtering. Guo et al. (2009) use the same filtering method as we use here, because they work with observational data that is only available as daily samples, this, we think offers one advantage of the 24-hour differencing method. Another advantage is that the 24-hour difference algorithm could be coded into GCMs in a manner that would allow the models to calculate the storm tracks online, to create climatological statistics, without saving a large amount of high-frequency temporal data.

12. *Section 2.4.5: I'm confused about the wave activity flux vectors. Shouldn't these be calculated for high-frequency eddies only since they characterize their influence on low-frequency blocking? e.g., Hoskins et al. 1983 JAS Fig. 15. Here the quantities used to calculate the fluxes are low pass filtered.*

[Ans] Hoskins et al. 1983 JAS formulates a quantity designated as the E-Vector. It can be thought of as the effective easterly momentum flux, where converging E-Vectors corresponds to a suppression of westerly mean flow, and thus the negative forcing of the eddies on the mean state. Hoskins et al. presents the E-vector for both low frequency (7-day lowpass) and high-frequency (7-day high pass) eddies computed with respect to the climatological mean.

In this work, the wave activity flux formulated by Takaya and Nakamura 2001 is utilized. In Takaya and Nakamura 2001 and Nakamura et al. 1997, wave activity fluxes are calculated as 8 day low-pass filtered eddies with the climatologies of the relevant input fields removed. The wave activity flux also relates eddy feedback onto the mean state, but by definition, is the pseudo-momentum associated with Rossby Waves. Both the E-vector and wave activity flux have proven to be useful, and the differences are subtle, but one advantage of the wave activity flux is that it is an instantaneous quantity.

In the original manuscript analysis, the stationary term of the wave activity flux was computed using 3 to 30 day bandpass, and 30-day lowpass filters to calculate the eddy and mean states, respectively. The formulation of wave activity flux in Takaya and Nakamura 2001, however, includes a non-stationary term that contributes much more in the high frequency regime. Therefore, to minimize the non-stationary influence of wave

activity flux, our analysis now instead focuses on wave activity fluxes of low frequency eddies calculated using an 8 to 30 day bandpass on the input fields. We have updated Section 2.4.3, lines 184-190, to be clearer:

"To better characterize the dynamical evolution of blocks within each model, wave activity flux vectors (hereinafter, $\vec{W}$) are calculated as described by Takaya and Nakamura (2001), hereinafter TN01. The wave activity flux relates eddy feedback onto the mean state and is essentially the pseudo-momentum associated with Rossby waves. Convergence of $\vec{W}$ is associated with blocking and an overall slowing or reversal of westerly flow. The formulation of $\vec{W}$ in TN01, includes a stationary term that dominates for quasi-stationary, low frequency eddies (i.e. 8- to 30-day timescales), and a non-stationary, group-velocity dependent term that is more relevant for higher frequency eddies. Here we calculate only the stationary, horizontal component of $\vec{W}$, and focus on contributions solely from the low frequency eddies."

13. *Lines 247-248: I suspect that the lower statistical significance threshold was used because the blocking statistics are not that different between the zonally symmetric versus asymmetric experiments. This supports general comment 3 above.*

[Ans] We received similar feedback in the other reviews. After further self-clarification we have now chosen a 95% confidence interval and more careful wording to describe quantitative differences throughout our analyses. See the methods subsection 2.5.4, lines 256-257:

"A 95% confidence interval is imposed as the significance threshold for all significance testing."

14. *Lines 290-291: I disagree. The contours differ by 25m, e.g. 275 versus 300.*

[Ans] This analysis is now removed.

15. *Lines 505-506: I believe Hassanzadeh et al. 2014 used a dry-dynamical core not an aquaplanet model.*

[Ans] We have updated any reference to this work to not refer to it as using an aquaplanet. Instead we use "idealized model with zonally symmetric forcing".

16. *Lines 537 and 545: Resonance has a very specific meaning, e.g., multiple reflection of waves on turning points following linear theory. I don't think it is what is implied here.*

[Ans] Agreed, this is removed.

17. *Figs 2,3,4 and 9 and related analysis: I suggest the authors interpret the wave activity fluxes with regards to flux convergence not the flux itself since this is the key dynamical quantity for blocking (Hoskins et al. 1983 JAS, Nakamura et al. 2018 science).*

[Ans] Agreed, this is now presented in Fig. 3 of the revised manuscript.

18. *Figs. 3 and related analysis: I suggest the authors compare the zonally-symmetric and asymmetric model simulations with reanalysis explicitly rather than reference previous work. Specifically, I suggest replacing Fig. 3 with a 3 x 3 panelled figure showing midlatitude blocking for reanalysis (top), zonally symmetric model (middle) and one zonally asymmetric model simulation for all 3 lifecycle stages (left, middle, right). This would also show that the two model configuration show similar results.*

[Ans] Agreed, see Fig. 3 and section 3.1.

**Response to Reviewer 3**

*The authors have used an idealized moist GCM and investigated some of the spatial and temporal characteristics of blocking events in the absence and in the presence of topography. I find the objectives of the paper and its results interesting and important (although further clarifications are needed). The paper is well structured and well written. I have a number of major and minor comments, which are listed below.*

[Ans] Thank you for the feedback. As discussed in the cover letter we have adjusted the article to focus more on the results from the aquaplanet, comparing them to results from reanalysis and idealized model integrations with topography.

*Recommendation: major revision*
*Major comments My major concern is that the paper is focused on too many questions, which have made the answers sometimes a bit too speculative. It appears to me that the three main questions are*

1- *Do the blocking events in aquaplanet simulations have the same dynamics as those of the real blocking events? This is a great question and its answer has important implications for our understanding of the dynamics of the blocking events, as for example, some blocking theories require zonal asymmetries in boundary conditions/forcings. The studies of Hu et al. (2008 GRL), Hassanzadeh et al. (2014 GRL), and more recently Nabizadeh et al. (2019 GRL) have shown the existence of blocking events in aquaplanet simulations and report some of their characteristics, but certainly, there is a need for further investigation, and I am glad that these authors have focused on this question. Given the importance of the answer, I believe that the statement in Lines 296-298 needs more support. To start, I suggest that you show the analysis of Fig. 3 for the ERA data as well, so that the readers can see the comparison side by side (rather than being referred to other papers such as TN01).*

[Ans] This feedback led to a greater emphasis on the aquaplanet results and the reformulation of research question 1 (see last bullet point of Major Changes section of this document). We now include citations for Nabizadeh et al. 2019 GRL anywhere we discuss previous results from idealized models with zonally symmetric forcing. Fig. 3 now shows a side by side comparison of the dynamical evolution of blocking events in the aquaplanet, topographic configurations, and reanalysis. Reviewer 2 also had similar thoughts

2- *Do the high-latitude blocks have the same dynamics as those of the midlatitude blocking events? The discussion in lines 286-292 is too speculative. I suggest that you show the analysis of Fig. 3 but for high latitude blocks (rather than the single panel in Fig. 4). Regarding the difference in dynamics: given the lack of W and weakness of the anomalies (pointed out in lines 290-291), is it possible that the high-latitude blocks are just cut-off highs that appear stationary because the zonal wind in the high latitudes is weak? (so there is really no maintenance mechanism?) What is the time scale of zonal advection in the high latitudes of the models (and what is it in the midlatitudes?)*

[Ans] To avoid issues and ambiguities related to midlatitude vs. high-latitude blocking, and based on a comment from reviewer 2, we have chosen to remove this section entirely from the manuscript. Perhaps this will be a focus of future work.

3- *What is the effect of topography on the duration, distribution, and dynamics? I think here the most interesting analysis is the comparison between Fig. 3 and 9. Whether the life cycle and dynamics are affected by the topography or not is an important question but is barely explored. I suggest that you further elaborate on these results. Otherwise, given the very idealized nature of topography here, I am not sure how much we can learn from the distribution and duration of different simulations with different topography configurations.*

[Ans] As mentioned above, Fig. 3 now includes a side-by-side comparison of blocks in the aquaplanet with blocks from the topographic configurations. The result remains the same.

With regards to duration, the revised manuscript provides better framing for the duration analysis, particularly in Section 3.2.4, lines 402-404 leading into the block duration analysis:

"The TwoMtn configuration has a greater hemispherically averaged blocking frequency than the other configurations (Table 2). This is despite the TwoMtn configuration having a lower total number of blocks than the 3 and 4 km SingleMtn configurations, respectively – meaning the blocks have a longer average duration in the 2-mountain configuration."

We still find the suggested increase in block duration for blocks forming near topography to be an interesting piece of the story. A natural question from the climatology analysis is: Do more events or longer lasting blocks cause the overall increase in hemispherically averaged blocking statistics within the idealized model integrations with topography compared to the aquaplanet? These results provide insight into this question, showing that it is a complex mixture of both. Differences in duration found in this study, albeit sometimes modest, also are consistent with popular theories linking a high propensity of blocking to weak zonal background flow (i.e. Nakamura and Huang 2018 science).

*Minor comments*
*L186: W is given in : : :..*

[Ans] Typo corrected, colon corrected.

*L247: 85% is too low. I suggest using a 95% confidence interval.*

[Ans] We received similar feedback in the other reviews. After further self-clarification we have now chosen a 95% confidence interval and more careful wording to describe quantitative differences throughout our analyses. See the methods subsection 2.5.4, lines 256-257:

"A 95% confidence interval is imposed as the significance threshold for all significance testing."

---

## Author Comment (AC5) · 8 May 2020

Please see attached for the response to all reviewers.

Page 1-2 contains general comments responding to all reviewers. Pages 3-5 refer to comments specifically from reviewer 1, pages 6-13 for reviewer 2, and 14-16 for reviewer 3.

We are asked to not submit the revised manuscript here, but it is ready to be adjusted for the next stages of review.

Thank you to the reviewers for all your feedback.

Please also note the supplement to this comment:

[Figure]

http://www.weather-clim-dynam-discuss.net/wcd-2020-2/wcd-2020-2-AC5-supplement.pdf
* * *

---

## Referee Report (RR1)

**REVIEW: Atmospheric Blocking in an Aquaplanet and the Impact of Orography by Narinesingh et al.**

**Summary:**

Scientific significance: good
Scientific quality: good
Presentation quality: good

I would like to thank the authors for considering my earlier comments. The manuscript is now more focused on the key questions that can be answered using the experiments and better uses previous work to explain the changes. Specially, the authors show that orography impacts the regional and hemispherically integrated statistics of blocking. The results could potentially explain differences in Northern versus Southern Hemisphere blocking and regional differences in Northern-Hemisphere blocking in reanalysis. Given the improvements, I recommend minor revisions before this paper can be published.

**General comments:**

1. Significance testing (section 2.5.4 and results): As I understand it, the authors use t-tests where the samples are taken from winter mean blocking frequencies. The t-test assumes the data are normally distributed, however, daily blocking events follow more of a Poisson distribution. Does averaging the events over one year produce normally distributed samples, either regionally or hemispherically integrated, as expected from the central limit theorem? I think it is important to confirm whether the statistical test is appropriate given that the changes in blocking statistics are modest and have large internal variability. Furthermore, have the authors tested whether the Northern versus Southern hemisphere statistics in reanalysis are statistically significant (L350-351)? Also I might be mistaken but I don't think the statistical significance mentioned on L493-494 was stated explicitly in the results section. I only found a mention of the regional blocking frequencies being significantly different on L385-390.

2. Results L279-282 and Fig. 3: I'm confused why the authors show Southern Hemisphere reanalysis results in Fig. 3a-c. I thought the idea behind the orography experiments was to mimic the configuration in the Northern-Hemisphere? I don't understand why the authors avoided 'regional variations' in Northern-Hemisphere blocking. Are the results different if Northern-Hemisphere blocks are shown? Furthermore, I'm confused why blocking events 'near the high-pressure anomaly of stationary waves' from the 3km mountain experiment was chosen. Are the results different if blocks from other regions are shown? If the authors need to be selective about which blocks to compare in the model and reanalysis, it suggests that the answer to question 1 in the Introduction is no.

3. Discussion and conclusion section: The results show modest changes in blocking statistics when comparing individual experiments to their control simulation and some of these changes are not statistically significant. The authors chose to emphasise the differences with the control simulation rather than the similarities. While I understand the reasoning for this choice, I think

the authors should also discuss the implications of the similarities, which are larger than the differences, which were mentioned in my earlier review. In particular, it could be that the model fails to capture the real effect of topography in reanalysis or that other processes, not included in the model, better explain the statistics in reanalysis.

**Specific comments:**

1. Abstract: I suggest including a sentence that states the knowledge gap in the literature to entice the reader. Its not clear why your results are important based on the abstract only.

2. L72-74: Questions 1 and 4 are similar. I suggest combining them. Also I suggest changing 'overall' in question 3 with 'hemispherically integrated' to be more concise.

3. L260 and other subsection titles: suggest using more descriptive titles. For example: 'Lifecycle of blocking in reanalysis and the idealized model' for section 3.1.

4. L296: Needs more explanation if the Danielson et al. 2005 citation is included.

5. L307-308: Citing the Woollings paper in the Introduction would be helpful to readers.

6. L395: For comparison, how high are the Rockies and Himalayas?

7. 418-419: Alternate explanation: other processes not included in the model could explain the differences.

8. L437-439: Instead of 'cautious suggestions', a more robust statement could be made that the differences are likely due to internal variability. The non-linear changes in duration in response to linear changes in topography support this interpretation.

9. L459-460: I don't understand what the citation is referring to here.

10. L475 'purely through eddy-eddy interactions': You haven't shown this. I would simply say that that blocks can be produced using a zonally symmetric model consistent with the role of eddy-eddy interaction generating blocks.

11. Conclusions and discussion: Since the authors have laid out specific questions in the Introduction, I think it would be useful for readers if you repeated them here before answering them.

12. Tables 2-3: It would help to have some measure of the year to year variability in these numbers to better understand the magnitude of the changes between experiments. For example, you could add 95% confidence intervals using +-2 standard deviations next to each value in the tables. You could also include a star symbol to denote which experiments are statistically different than the control.

13. Fig. 2 and 3: Suggest presenting both figures in the same format to better compare.

14. Caption in Fig. 4: I think this may be wrong.

15. Fig. 8: It is difficult to compare different experiments because they overlap. I suggest splitting this plot into 4 panels, each with the control and 1 experiment with the mean and the spread. Why is the two mountain experiment left out?

---

## Author Response (AR2)

To the editor and reviewer:

For this revision, we have addressed the new comments from Reviewers 2 and 3. Reviewer 3 found a few typos and otherwise found the manuscript to be well written and complete. We have made sure to fix the typos. Reviewer 2 had a few more questions and comments. What follows are our responses to Reviewer 2. The reviewer's comments are italicized in blue. Our responses are in black.

*REVIEW: Atmospheric Blocking in an Aquaplanet and the Impact of Orography by Narinesingh et al.*

*Summary:*
*Scientific significance: good*
*Scientific quality: good*
*Presentation quality: good*

*I would like to thank the authors for considering my earlier comments. The manuscript is now more focused on the key questions that can be answered using the experiments and better uses previous work to explain the changes. Specially, the authors show that orography impacts the regional and hemispherically integrated statistics of blocking. The results could potentially explain differences in Northern versus Southern Hemisphere blocking and regional differences in Northern-Hemisphere blocking in reanalysis. Given the improvements, I recommend minor revisions before this paper can be published.*

[Ans] Thank you for your feedback and helpful suggestions throughout the peer review process of this article. Please see below for specific responses to your comments.

**General comments:**

1. *Significance testing (section 2.5.4 and results): As I understand it, the authors use t-tests where the samples are taken from winter mean blocking frequencies. The t-test assumes the data are normally distributed, however, daily blocking events follow more of a Poisson distribution. Does averaging the events over one year produce normally distributed samples, either regionally or hemispherically integrated, as expected from the central limit theorem? I think it is important to confirm whether the statistical test is appropriate given that the changes in blocking statistics are modest and have large internal variability. Furthermore, have the authors tested whether the Northern versus Southern hemisphere statistics in reanalysis are statistically significant (L350-351)? Also I might be mistaken but I don't think the statistical significance mentioned on L493-494 was stated explicitly in the results section. I only found a mention of the regional blocking frequencies being significantly different on L385-390.*

   [Ans.] Thank you for pointing out the need to check the appropriateness of the t-test in our significance testing. In the figure below (Response Fig. 1), we take hemispherically averaged blocking frequency for each cool season and create histograms for each model configuration. Superimposed on these plots are the parameterized normal distributions using the mean and standard deviation.

For the 300-year aquaplanet, the histogram does indeed show behavior resembling a normal distribution in terms of shape and magnitude (response Fig. 1a). The other model configurations also show signs of convergence to a normal distribution. One caveat, however, is that the histograms for the orographic configurations (i.e. 30 years of data) appear to be under-sampled and noisy. Nevertheless, one strength of the t-test (Welch's t test in our analysis) is the test statistic is a function of both sample variances. Therefore, we deem this approach to be an appropriate method for the hemispherically averaged blocking frequency analysis.

On the other hand, if we focus on the distributions of average blocking frequency at a given gridpoint, rather than the hemispheric averages, we find that the distributions do not resemble a normal distribution. These, as you mentioned, more naturally fit Poisson distributions (not shown). Therefore, we instead use the Mann-Whitney u test for the significance testing of gridpoint-wise blocking frequency. One advantage of the u test is that it does not rely on parameterized fitting to any specific distribution. Figures 4e-h, 5b, and 9b are updated using the u test, and yield nearly identical results to the t test used before. The duration analysis also now uses as u test, which yields the same significance testing results as before. The methods section lines 250-265 are now updated to explain our choices in significance testing

Significance testing on hemispherically averaged cool season blocking frequency for the northern and southern hemispheres does indeed find that the Northern Hemisphere has significantly more blocking. Lines 375-376 now mentions this result. Though the absolute difference between both hemispheres is a modest 1.1% (2.7 % for the NH minus 1.6% for the SH), in terms of relative change, this is a 68% increase. The idealized model does not show such drastic changes from configuration to configuration, but when comparing the aquaplanet to the 3 km single mountain run for example, we still find a 15 % increase in hemispherically averaged blocking frequency. The reason for this discrepancy is discussed in greater detail in our response to your Major comment #3.

With regards to the significance results mentioned in the conclusion lines 493-494, we now explicitly mention this in the results section lines 415-416, and 429. Thank you for catching that.

[Figure]

Response Figure 1. Histograms of cool season hemispheric-average blocking frequency (blue bars) and parameterized normal distributions using the sample mean and standard deviation (red curves). (a) is for the aquaplanet idealized model integration, and (b)-(e) are for the single mountain integrations of varying height.

2. *Results L279-282 and Fig. 3: I'm confused why the authors show Southern Hemisphere reanalysis results in Fig. 3a-c. I thought the idea behind the orography experiments was to mimic the configuration in the Northern-Hemisphere? I don't understand why the authors avoided 'regional variations' in Northern-Hemisphere blocking. Are the results different if Northern-Hemisphere blocks are shown? Furthermore, I'm confused why blocking events 'near the high-pressure anomaly of stationary waves' from the 3km mountain experiment was chosen. Are the results different if blocks from other regions are shown? If the authors need to be selective about which blocks to compare in the model and reanalysis, it suggests that the answer to question 1 in the Introduction is no.*

    [Ans] We can see your point. We are not trying to pick and choose results, and we have adjusted the manuscript to make that point clear, as detailed below.

    Figure 3 has been remade and now it includes a composite for the NH – albeit the blocks over the oceans only. As the figure shows, the block-centered composites of geopotential height anomalies, wave activity flux vectors, and wave activity flux divergence NH and SH in reanalysis and the models share many characteristics.

Our reason for not including the NH in the previous iteration of this manuscript is as follows: (1) for the sake of comparing observations with the aquaplanet, it makes sense to use the SH. , and (2) Nakamura et al., 1997, MWR, highlights the fact that the North Pacific differs from the North Atlantic in the compositing. We found the same difference they discuss: for the North Pacific the wave activity flux into the block at the initialization phase is weaker than it is for the North Atlantic. However, we agree with your point: since we are making the case that the idealized model captures many of the features found in observations, we should show it for both the NH and SH. Therefore, the text discussing the compositing has been adjusted to account for all of these changes (section 3.1). We also include a discussion of the differences between the blocks of the ocean for the North Pacific and North Atlantic. This can be found in lines 296-297, and 301-304.

Your second question is regarding why, in the 3km single mountain case, we chose to show blocks near the high-pressure anomaly of the stationary wave rather for the entire model domain (these figures are now on the bottom row of the new Figure 3). Here, we admit that up until now, we did not fully appreciate the significance of this figure – so thank you for pressing on this one.

For the previous iteration of this manuscript, we chose to show blocks in the 3km single mountain configuration from the region near the high-pressure anomaly to emphasize that they evolve in a dynamically similar manner to blocks forced without the presence of a climatological stationary wave (i.e. as in the aquaplanet). However, your questions have helped us appreciate a key difference between the model and observations: in the model configured with mountains, the regional differences in the blocking composites are smaller than those found in observations. Thus, the model is missing some aspects of the dynamics that are associated with forcing of the blocks in the NH. That being said, the model does well to capture the geographical preference for blocking to occur in regions upstream of mountains. This, we think is a useful result: despite the model with mountains not being able to fully capture the internal physics of the blocks (as revealed by the composites being so similar for the model with mountains and aquaplanet), the model configured with mountains does capture the spatial shifts in the location of the blocks. Therefore, in the conclusion we have included some commentary on this result in lines 540-549:

"We note that the influence of mountains in our model is not identical to the differences between the NH and SH in observations. First, from the block-centered composites (Fig. 3), it was clear that the NH vs SH differences in observations for Z500 anomalies and wave activity flux are larger than those found for the aquaplanet as compared to the idealized configurations with orography. This is true for the case shown in Fig. 3 (3 km single mountain) and all other model configurations with orography. Additionally, the hemispherically-averaged blocking frequency in the NH is much larger than the SH as compared to the aquaplanet versus any model configuration with mountains. On the other hand, the spatial distribution of blocking minimizes at the storm track entrance and maximizes near the anticyclonic peak of the stationary wave, is exactly captured in our model. Thus, there are some similarities for our aquaplanet and orographic configurations inconsistent with reanalysis – which may be due to deficiencies in the model (discussed below), but there are also important differences when orography is added."

3.  *Discussion and conclusion section: The results show modest changes in blocking statistics when comparing individual experiments to their control simulation and some of these changes are not statistically significant. The authors chose to emphasise the differences with the control simulation rather than the similarities. While I understand the reasoning for this choice, I think the authors should also discuss the implications of the similarities, which are larger than the differences, which were mentioned in my earlier review. In particular, it could be that the model fails to capture the real effect of topography in reanalysis or that other processes, not included in the model, better explain the statistics in reanalysis.*

[Ans] We have addressed this comment by adding more commentary in the conclusion section (lines 550-563). We highlight the similarities, such as the hemispheric averages in blocking amount, but we also point out the differences in spatial distributions. We use this, plus the new composite results discussed in the response to your major comment #2 to highlight the following:

"the fact that the compositing did not show the same differences for aquaplanet vs. mountains cases as SH vs. NH implies that the subtleties of the block-centered compositing dynamics do not determine the spatial distribution of the blocks."

**Specific comments:**

1.  *Abstract: I suggest including a sentence that states the knowledge gap in the literature to entice the reader. Its not clear why your results are important based on the abstract only.*

    [Ans] We have added to the abstract to reflect this (lines 1, and 29-31). There we mention how our results help explain some of the differences in blocking between the NH and SH as well as the importance of stationary waves. We also mention how this study shows the utility of an idealized model for understanding blocks observed in reality (lines 20-21).

2.  *L72-74: Questions 1 and 4 are similar. I suggest combining them. Also I suggest changing 'overall' in question 3 with 'hemispherically integrated' to be more concise.*

    [Ans] We choose to keep questions 1 and 4 separate. Though they are linked, we find them to be distinct. Question 1 is focused on similarities in dynamical evolution of blocking between models, whereas question 4 focuses on duration. In question 2, as per your suggestion, we change "overall" to "hemispherically averaged".

3.  *L260 and other subsection titles: suggest using more descriptive titles. For example: 'Lifecycle of blocking in reanalysis and the idealized model' for section 3.1.*

    [Ans] We appreciate your suggestion but it is our preference to keep the headings as is with the exception of 3.1. For this subsection, we change the name to "Blocking in the aquaplanet, dynamical aspects and intermodel comparison" to be more descriptive.

4.  *L296: Needs more explanation if the Danielson et al. 2005 citation is included.*

    [Ans] We have added some context to this in lines 314-316.

5.  *L307-308: Citing the Woollings paper in the Introduction would be helpful to readers.*

    [Ans] Agreed, we now add a line in the beginning paragraph of the introduction (lines 36-37):

    "For readers looking for a comprehensive review of blocking see Woollings et al. 2018."

6.  *L395: For comparison, how high are the Rockies and Himalayas?*

    [Ans] We now include this in lines 421-423:

    "3 km height is meant to be semi-realistic; the values are lower than the maxima for the Rockies and the Tibetan Plateau (~4400 m and ~8800 m, respectively) – however the mountains are substantial enough to generate obvious changes in the circulation (as evidenced in the Single Mountain experiments)."

7.  *418-419: Alternate explanation: other processes not included in the model could explain the differences.*

    [Ans] Noted, we now include this caveat and others in the conclusion section lines 556-563.

8.  *L437-439: Instead of 'cautious suggestions', a more robust statement could be made that the differences are likely due to internal variability. The non-linear changes in duration in response to linear changes in topography support this interpretation.*

    [Ans] We leave the cautious suggestion part in but include your suggestion about internal variability. See lines 466-467.

9.  *L459-460: I don't understand what the citation is referring to here.*

    [Ans] Agreed, this was unclear. We have removed this. After some clarification, we also take out the incorrect statement that land in these experiments includes orographic drag. We now cite Pithan et al. 2016 in lines 556-558 of the conclusion instead.

10. *L475 'purely through eddy-eddy interactions': You haven't shown this. I would simply say that that blocks can be produced using a zonally symmetric model consistent with the role of eddy-eddy interaction generating blocks.*

    [Ans] Thank you for the suggestion. We now adjust the wording in lines 508-509 to reflect this.

11. *Conclusions and discussion: Since the authors have laid out specific questions in the Introduction, I think it would be useful for readers if you repeated them here before answering them.*

    [Ans] Agreed, we have included this reminder in the beginning of the conclusions section lines 502-507.

12. *Tables 2-3: It would help to have some measure of the year to year variability in these numbers to better understand the magnitude of the changes between experiments. For example, you could add 95% confidence intervals using +-2 standard deviations next to each value in the tables. You could also include a star symbol to denote which experiments are statistically different than the control.*

    [Ans] Thank you for the suggestion. We have now included the standard deviations in this table as well as asterisks denoting statistically different results from the aquaplanet.

13. *Fig. 2 and 3: Suggest presenting both figures in the same format to better compare.*

    [Ans] We choose to keep these figures as is because they provide two different, but equally useful ways of visualizing blocks. In Fig. 2 we see blocks as projected onto a sphere, whereas Fig. 3 shows blocking from an equal-area perspective.

14. *Caption in Fig. 4: I think this may be wrong.*

    [Ans] Thank you for catching this error. This caption has been corrected

15. *Fig. 8: It is difficult to compare different experiments because they overlap. I suggest splitting this plot into 4 panels, each with the control and 1 experiment with the mean and the spread. Why is the two mountain experiment left out?*

This figure has been updated as per your suggestion and now includes the two mountain experiment. The two mountain experiment was originally left out to eliminate clutter, but this format is much cleaner.

[revised manuscript text omitted]

---

## Author Response (AR3)

To the co-editor:

Thank you for your facilitation of the review process of this paper. We appreciate your suggestions to better enhance the clarity of this article. Below you will see specific responses to your technical corrections, followed by a marked-up revised version of the manuscript's current iteration.

*L253ff. I think it would help the reader understanding why you choose a u-test if it would be stated explicitly that the gridpointwise blocking statistics approximately follows a Poisson distribution. This in particular concerning the distinction to normally distributed hemispheric blocking statistics.*
[Ans] Explicit statement of this is now contained in lines 250-251.

*L345: Delete «for the».*
[Ans] Corrected

*L502ff: It is not necessary to explicitly repeat the questions here, but rather just say that here you present the answers to the specific questions raised at the end of the introduction. Furthermore, it would help, if the question the subsequently listed results pertain to, would be higlighted. For instance, «There is an increase in block duration for blocks originating near mountains, though the statistics are not robust (Question 4).»*
[Ans] This explicit reiteration of the original research questions is now removed, and a transition sentence is added (line 501). Reference to each respective research question is now included in lines 517-524.

*L544: «… than in the SH ..»*
*L545 – L549: The two sentences need rephrasing. In particular, it is not clear what «… , is exactly captured in our model» refers to. In addition, the subsequent sentence does not seem grammatically correct. Do you mean «…, there are some similarities between our aquaplanet and orographic configurations that are inconsistent with reanalysis...»?*
*L546: «Spatial distribution of blocking» is unclear, perhaps you mean «frequency of blocking»?*
[Ans] The above are all fixed and addressed in the revised version of this paragraph. See lines 535-545.

*Caption Figure 3: remove references to rows, just stating «(a, e, i) ...» is enough.*
[Ans] Corrected

[revised manuscript text omitted]